# SNIPS: Solving Noisy Inverse Problems Stochastically

**Bahjat Kawar, Gregory Vaksman, Michael Elad**
Computer Science Department, Technion, Haifa, Israel
`{bahjat.kawar, grishav, elad}@cs.technion.ac.il`

## Abstract

In this work we introduce a novel stochastic algorithm dubbed *SNIPS*, which draws samples from the posterior distribution of any linear inverse problem, where the observation is assumed to be contaminated by additive white Gaussian noise. Our solution incorporates ideas from Langevin dynamics and Newton's method, and exploits a pre-trained minimum mean squared error (MMSE) Gaussian denoiser. The proposed approach relies on an intricate derivation of the posterior score function that includes a singular value decomposition (SVD) of the degradation operator, in order to obtain a tractable iterative algorithm for the desired sampling. Due to its stochasticity, the algorithm can produce multiple high perceptual quality samples for the same noisy observation. We demonstrate the abilities of the proposed paradigm for image deblurring, super-resolution, and compressive sensing. We show that the samples produced are sharp, detailed and consistent with the given measurements, and their diversity exposes the inherent uncertainty in the inverse problem being solved.

## 1 Introduction

Many problems in the field of image processing can be cast as noisy linear inverse problems. This family of tasks includes denoising, inpainting, deblurring, super resolution, compressive sensing, and many other image recovery problems. A general linear inverse problem is posed as

$$\mathbf{y} = \mathbf{H}\mathbf{x} + \mathbf{z}, \tag{1}$$

where we aim to recover a signal $\mathbf{x}$ from its measurement $\mathbf{y}$, given through a linear degradation operator $\mathbf{H}$ and a contaminating noise, being additive, white and Gaussian, $\mathbf{z} \sim \mathcal{N}\left(0, \sigma_0^2 \mathbf{I}\right)$. In this work we assume that both $\mathbf{H}$ and $\sigma_0$ are known.

Over the years, many strategies, algorithms and underlying statistical models were developed for handling image restoration problems. A key ingredient in many of the classic attempts is the prior that aims to regularize the inversion process and lead to visually pleasing results. Among the various options explored, we mention sparsity-inspired techniques [13, 55, 11], local Gaussian-mixture modeling [57, 63], and methods relying on non-local self-similarity [6, 9, 36, 51]. More recently, and with the emergence of deep learning techniques, a direct design of the recovery path from $\mathbf{y}$ to an estimate of $\mathbf{x}$ took the lead, yielding state-of-the-art results in various linear inverse problems, such as denoising [25, 59, 61, 52], deblurring [22, 48], super resolution [10, 17, 54] and other tasks [29, 28, 19, 16, 37, 58].

Despite the evident success of the above techniques, many image restoration algorithms still have a critical shortcoming: In cases of severe degradation, most recovery algorithms tend to produce washed out reconstructions that lack details. Indeed, most image restoration techniques seek a reconstruction that minimizes the mean squared error between the restored image, $\hat{\mathbf{x}}$, and the unknown original one, $\mathbf{x}$. When the degradation is acute and information is irreversibly lost, image reconstruction becomes a highly ill-posed problem, implying that many possible clean images could explain the given

35th Conference on Neural Information Processing Systems (NeurIPS 2021).

measurements. The MMSE solution averages all these candidate solutions, being the conditional mean of the posterior of $\mathbf{x}$ given $\mathbf{y}$, leading to an image with loss of fine details in the majority of practical cases. A recent work reported in [5] has shown that reconstruction algorithms necessarily suffer from a perception-distortion tradeoff, *i.e.*, targeting a minimization of the error between $\hat{\mathbf{x}}$ and $\mathbf{x}$ (in any metric) is necessarily accompanied by a compromised perceptual quality. As a consequence, as long as we stick to the tendency to design recovery algorithms that aim for minimum MSE (or other distances), only a limited perceptual improvement can be expected.

When perceptual quality becomes our prime objective, the strategy for solving inverse problems must necessarily change. More specifically, the solution should concentrate on producing a sample (or many of them) from the posterior distribution $p(\mathbf{x}|\mathbf{y})$ instead of its conditional mean. Recently, two such approaches have been suggested – GAN-based and Langevin sampling. Generative Adversarial Networks (GANs) have shown impressive results in generating realistically looking images (*e.g.*, [14, 35]). GANs can be utilized for solving inverse problems while producing high-quality images (see *e.g.* [2, 31, 34]). These solvers aim to produce a diverse set of output images that are consistent with the measurements, while also being aligned with the distribution of clean examples. A major disadvantage of GAN-based algorithms for inverse problems is their tendency (as practiced in [2, 31, 34]) to assume noiseless measurements, a condition seldom met in practice. An exception to this is the work reported in [33], which adapts a conditional GAN to become a stochastic denoiser.

The second approach for sampling from the posterior, and the one we shall be focusing on in this paper, is based on Langevin dynamics. This core iterative technique enables sampling from a given distribution by leveraging the availability of the score function – the gradient of the log of the probability density function [38, 3]. The work reported in [44, 20, 46] utilizes the annealed Langevin dynamics method, both for image synthesis and for solving *noiseless* inverse problems.[1] Their synthesis algorithm relies on an MMSE Gaussian denoiser (given as a neural network) for approximating a gradually blurred score function. In their treatment of inverse problems, the conditional score remains tractable and manageable due to the noiseless measurements assumption.

The question addressed in this paper is the following: How can the above line of Langevin-based work be generalized for handling linear inverse problems, as in Equation 1, in which the measurements are noisy? A partial and limited answer to this question has already been given in [21] for the tasks of image denoising and inpainting. The present work generalizes these ([44, 20, 46, 21]) results, and introduces a systematic way for sampling from the posterior distribution of any given noisy linear inverse problem. As we carefully show, this extension is far from being trivial, due to two prime reasons: (i) The involvement of the degradation operator $\mathbf{H}$, which poses a difficulty for establishing a relationship between the reconstructed image and the noisy observation; and (ii) The intricate connection between the measurements' and the synthetic annealed Langevin noise. Our proposed remedy is a decorrelation of the measurements equation via a singular value decomposition (SVD) of the operator $\mathbf{H}$, which decouples the dependencies between the measurements, enabling each to be addressed by an adapted iterative process. In addition, we define the annealing noise to be built as portions of the measurement noise itself, in a manner that facilitates a constructive derivation of the conditional score function.

Following earlier work [44, 20, 46, 21], our algorithm is initialized with a random noise image, gradually converging to the reconstructed result, while following the direction of the log-posterior gradient, estimated using an MMSE denoiser. Via a careful construction of the gradual annealing noise sequence, from very high values to low ones, the entries in the derived score switch mode. Those referring to non-zero singular values start by being purely dependent on the measurements, and then transition to incorporate prior information based on the denoiser. As for entries referring to zero singular values, their corresponding entries undergo a pure synthesis process based on the prior-only score function. Note that the denoiser blends values in the evolving sample, thus intermixing the influence of the gradient entries. Our derivations include an analytical expression for a position-dependent step size vector, drawing inspiration from Newton's method in optimization. This stabilizes the algorithm and is shown to be essential for its success.

We refer hereafter to our algorithm as *SNIPS* (Solution of Noisy Inverse Problems Stochastically). Observe that as we target to sample from the posterior distribution $p(\mathbf{x}|\mathbf{y})$, different runs of SNIPS on the same input necessarily yield different results, all of which valid solutions to the given inverse

---

[1]The work reported in [18, 42, 26] and [15, 23] is also relevant to this discussion, but somewhat different. We shall specifically address these papers' content and its relation to our work in section 2.

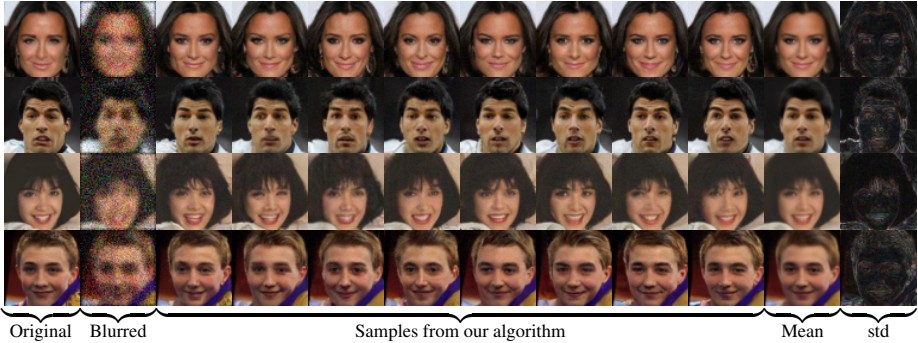

Original  Blurred  Samples from our algorithm  Mean  std

Figure 1: Deblurring results on CelebA [27] images (uniform $5 \times 5$ blur and an additive noise with $\sigma_0 = 0.1$). Here and in all other shown figures, the standard deviation image is scaled by 4 for better visual inspection.

problem. This should not come as a surprise, as ill-posedness implies that there are multiple viable solutions for the same data, as has already been suggested in the context of super resolution [31, 2, 34]. We demonstrate SNIPS on image deblurring, single image super resolution, and compressive sensing, all of which contain non-negligible noise, and emphasize the high perceptual quality of the results, their diversity, and their relation to the MMSE estimate.

To summarize, this paper's contributions are threefold:

- We present an intricate derivation of the blurred posterior score function for general noisy inverse problems, where both the measurement and the target image contain delicately inter-connected additive white Gaussian noise.

- We introduce a novel stochastic algorithm – *SNIPS* – that can sample from the posterior distribution of these problems. The algorithm relies on the availability of an MMSE denoiser.

- We demonstrate impressive results of *SNIPS* on image deblurring, single image super resolution, and compressive sensing, all of which are highly noisy and ill-posed.

Before diving into the details of this work, we should mention that using Gaussian denoisers iteratively for handling general linear inverse problems has been already proposed in the context of the Plug-and-Play-Prior (PnP) method [53] and RED [39], and their many followup papers (*e.g.*, [60, 30, 1, 49, 7, 50, 40, 4]). However, both PnP and RED are quite different from our work, as they do not target sampling from the posterior, but rather focus on MAP or MMSE estimation.

## 2   Background

The Langevin dynamics algorithm [3, 38] suggests sampling from a probability distribution $p(\mathbf{x})$ using the iterative transition rule

$$\mathbf{x}_{t+1} = \mathbf{x}_t + \alpha \nabla_{\mathbf{x}_t} \log p(\mathbf{x}_t) + \sqrt{2\alpha} \mathbf{z}_t \; , \qquad (2)$$

where $\mathbf{z}_t \sim \mathcal{N}(0, \mathbf{I})$ and $\alpha$ is an appropriately chosen small constant. The added $\mathbf{z}_t$ allows for stochastic sampling, avoiding a collapse to a maximum of the distribution. Initialized randomly, after a sufficiently large number of iterations, and under some mild conditions, this process converges to a sample from the desired distribution $p(\mathbf{x})$ [38].

The work reported in [44] extends the aforementioned algorithm into *annealed Langevin dynamics*. The annealing proposed replaces the score function in Equation 2 with a blurred version of it, $\nabla_{\tilde{\mathbf{x}}_t} \log p(\tilde{\mathbf{x}}_t)$, where $\tilde{\mathbf{x}}_\mathbf{t} = \mathbf{x}_t + \mathbf{n}$ and $\mathbf{n} \sim \mathcal{N}(0, \sigma^2 \mathbf{I})$ is a synthetically injected noise. The core idea is to start with a very high noise level $\sigma$ and gradually drop it to near-zero, all while using a step size $\alpha$ dependent on the noise level. These changes allow the algorithm to converge much faster and perform better, because it widens the basin of attraction of the sampling process. The work in [20] further develops this line of work by leveraging a brilliant relation attributed to Miyasawa [32] (also known as Stein's integration by parts trick [47] or Tweedie's identity [12]). It is given as

$$\nabla_{\tilde{\mathbf{x}}_t} \log p(\tilde{\mathbf{x}}_t) = \frac{\mathbf{D}(\tilde{\mathbf{x}}_t, \sigma) - \tilde{\mathbf{x}}_t}{\sigma^2}, \qquad (3)$$

where $\mathbf{D}\left(\tilde{\mathbf{x}}_t, \sigma\right) = \mathbb{E}\left[\mathbf{x}|\tilde{\mathbf{x}}_t\right]$ is the minimizer of the MSE measure $\mathbb{E}\left[\|\mathbf{x} - \mathbf{D}\left(\tilde{\mathbf{x}}_t, \sigma\right)\|_2^2\right]$, which can be approximated using a denoising neural network. This facilitates the use of denoisers in Langevin dynamics as a replacement for the evasive score function.

When turning to solve inverse problems, previous work suggests sampling from the posterior distribution $p\left(\mathbf{x}|\mathbf{y}\right)$ using annealed Langevin dynamics [20, 46, 21] or similar methods [15, 18, 42, 26], by replacing the score function used in the generation algorithm with a conditional one. As it turns out, if limiting assumptions can be posed on the measurements formation, the conditional score is tractable, and thus generalization of the annealed Langevin process to these problems is within reach. Indeed, in [44, 20, 46, 42, 26] the core assumption is $\mathbf{y} = \mathbf{Hx}$ for specific and simplified choices of $\mathbf{H}$ and with no noise in the measurements. The works in [15, 23] avoid these difficulties altogether by returning to the original (non-annealed) Langevin method, with the unavoidable cost of becoming extremely slow. In addition, their algorithms are demonstrated on inverse problems in which the additive noise is restricted to be very weak. The work in [21] is broader, allowing for an arbitrary additive white Gaussian noise, but limits $\mathbf{H}$ to the problems of denoising or inpainting. While all these works demonstrate high quality results, there is currently no clear way for deriving the blurred score function of a general linear inverse problem as posed in Equation 1. In the following, we present such a derivation.

## 3 The Proposed Approach: Deriving the Conditional Score Function

### 3.1 Problem Setting

We consider the problem of recovering a signal $\mathbf{x} \in \mathbb{R}^N$ (where $\mathbf{x} \sim p\left(\mathbf{x}\right)$ and $p\left(\mathbf{x}\right)$ is unknown) from the observation $\mathbf{y} = \mathbf{Hx} + \mathbf{z}$, where $\mathbf{y} \in \mathbb{R}^M, \mathbf{H} \in \mathbb{R}^{M \times N}, M \le N, \mathbf{z} \sim \mathcal{N}\left(0, \sigma_0^2\mathbf{I}\right)$, and $\mathbf{H}$ and $\sigma_0$ are known.[2] Our ultimate goal is to sample from the posterior $p\left(\mathbf{x}|\mathbf{y}\right)$. However, since access to the score function $\nabla_{\mathbf{x}} \log p(\mathbf{x}|\mathbf{y})$ is not available, we retarget our goal, as explained above, to sampling from blurred posterior distributions, $p\left(\tilde{\mathbf{x}}|\mathbf{y}\right)$, where $\tilde{\mathbf{x}} = \mathbf{x} + \mathbf{n}$ and $\mathbf{n} \sim \mathcal{N}\left(0, \sigma^2\mathbf{I}\right)$, with noise levels $\sigma$ starting very high, and decreasing towards near-zero.

As explained in the supplemental material, the sampling should be performed in the SVD domain in order to get a tractable derivation of the blurred score function. Thus, we consider the singular value decomposition (SVD) of $\mathbf{H}$, given as $\mathbf{H} = \mathbf{U}\boldsymbol{\Sigma}\mathbf{V}^T$, where $\mathbf{U} \in \mathbb{R}^{M \times M}$ and $\mathbf{V} \in \mathbb{R}^{N \times N}$ are orthogonal matrices, and $\boldsymbol{\Sigma} \in \mathbb{R}^{M \times N}$ is a rectangular diagonal matrix containing the singular values of $\mathbf{H}$, denoted as $\{s_j\}_{j=1}^M$ in descending order ($s_1 > s_2 > \cdots > s_{M-1} > s_M \ge 0$). For convenience of notations, we also define $s_j = 0$ for $j = M+1, \ldots, N$. To that end, we notice that

$$p\left(\tilde{\mathbf{x}}|\mathbf{y}\right) = p\left(\tilde{\mathbf{x}}|\mathbf{U}^T\mathbf{y}\right) = p\left(\mathbf{V}^T\tilde{\mathbf{x}}|\mathbf{U}^T\mathbf{y}\right). \tag{4}$$

The first equality holds because the multiplication of $\mathbf{y}$ by the orthogonal matrix $\mathbf{U}^T$ does not add or remove information, and the second equality holds because the multiplication of $\tilde{\mathbf{x}}$ by $\mathbf{V}^T$ does not change its probability [24]. Therefore, sampling from $p\left(\mathbf{V}^T\tilde{\mathbf{x}}|\mathbf{U}^T\mathbf{y}\right)$ and then multiplying the result by $\mathbf{V}$ will produce the desired sample from $p\left(\tilde{\mathbf{x}}|\mathbf{y}\right)$. As we are using Langevin dynamics, we need to calculate the conditional score function $\nabla_{\mathbf{V}^T\tilde{\mathbf{x}}} \log p\left(\mathbf{V}^T\tilde{\mathbf{x}}|\mathbf{U}^T\mathbf{y}\right)$. For simplicity, we denote hereafter $\mathbf{y}_T = \mathbf{U}^T\mathbf{y}$, $\mathbf{z}_T = \mathbf{U}^T\mathbf{z}$, $\mathbf{x}_T = \mathbf{V}^T\mathbf{x}$, $\mathbf{n}_T = \boldsymbol{\Sigma}\mathbf{V}^T\mathbf{n}$, and $\tilde{\mathbf{x}}_T = \mathbf{V}^T\tilde{\mathbf{x}}$. Observe that with these notations, the measurements equation becomes

$$\mathbf{y} = \mathbf{Hx} + \mathbf{z} = \mathbf{U}\boldsymbol{\Sigma}\mathbf{V}^T\mathbf{x} + \mathbf{z},$$

and thus

$$\mathbf{U}^T\mathbf{y} = \boldsymbol{\Sigma}\mathbf{V}^T\mathbf{x} + \mathbf{U}^T\mathbf{z} = \boldsymbol{\Sigma}\mathbf{V}^T(\tilde{\mathbf{x}} - \mathbf{n}) + \mathbf{U}^T\mathbf{z} = \boldsymbol{\Sigma}\mathbf{V}^T\tilde{\mathbf{x}} - \boldsymbol{\Sigma}\mathbf{V}^T\mathbf{n} + \mathbf{U}^T\mathbf{z},$$

where we have relied on the relation $\tilde{\mathbf{x}} = \mathbf{x} + \mathbf{n}$. This leads to

$$\mathbf{y}_T = \boldsymbol{\Sigma}\tilde{\mathbf{x}}_T - \mathbf{n}_T + \mathbf{z}_T. \tag{5}$$

In this formulation, which will aid in deriving the conditional score, our aim is to make design choices on $\mathbf{n}_T$ such that $\mathbf{z}_T - \mathbf{n}_T$ has uncorrelated entries and is independent of $\tilde{\mathbf{x}}_T$. This brings us to the formation of the synthetic annealed noise, which is an intricate ingredient in our derivations.

---

[2]We assume $M \le N$ for ease of notations, and because this is the common case. However, the proposed approach and all our derivations work just as well for $M > N$.

We base this formation on the definition of a sequence of noise levels $\{\sigma_i\}_{i=1}^{L+1}$ such that $\sigma_1 > \sigma_2 > \cdots > \sigma_L > \sigma_{L+1} = 0$, where $\sigma_1$ is high (possibly $\sigma_1 > \|\mathbf{x}\|_\infty$) and $\sigma_L$ is close to zero. We require that for every $j$ such that $s_j \neq 0$, there exists $i_j$ such that $\sigma_{i_j} s_j < \sigma_0$ and $\sigma_{i_j-1} s_j > \sigma_0$. This implies $\forall i : \sigma_i s_j \neq \sigma_0$, which helps ease notations. SNIPS works just as well for $\sigma_i s_j = \sigma_0$.

Using $\{\sigma_i\}_{i=1}^{L+1}$, we would like to define $\{\tilde{\mathbf{x}}_i\}_{i=1}^{L+1}$, a sequence of noisy versions of $\mathbf{x}$, where the noise level in $\tilde{\mathbf{x}}_i$ is $\sigma_i$. One might be tempted to define these noise additions as independent of the measurement noise $\mathbf{z}$. However, this option leads to a conditional score term that cannot be calculated analytically, as explained in the supplemental material. Therefore, we define these noise additions differently, as carved from $\mathbf{z}$ in a gradual fashion. To that end, we define $\tilde{\mathbf{x}}_{L+1} = \mathbf{x}$, and for every $i = L, L-1, \ldots, 1$: $\tilde{\mathbf{x}}_i = \tilde{\mathbf{x}}_{i+1} + \boldsymbol{\eta}_i$, where $\boldsymbol{\eta}_i \sim \mathcal{N}\left(0, \left(\sigma_i^2 - \sigma_{i+1}^2\right)\mathbf{I}\right)$. This results in $\tilde{\mathbf{x}}_i = \mathbf{x} + \mathbf{n}_i$, where $\mathbf{n}_i = \sum_{k=i}^{L} \boldsymbol{\eta}_k \sim \mathcal{N}\left(0, \sigma_i^2\mathbf{I}\right)$.

And now we turn to define the statistical dependencies between the measurements' noise $\mathbf{z}$ and the artificial noise vectors $\boldsymbol{\eta}_i$. Since $\boldsymbol{\eta}_i$ and $\mathbf{z}$ are each Gaussian with uncorrelated entries, so are the components of the vectors $\boldsymbol{\Sigma}\mathbf{V}^T\boldsymbol{\eta}_i$, $\boldsymbol{\Sigma}\mathbf{V}^T\mathbf{n}_i$, and $\mathbf{z}_T$. In order to proceed while easing notations, let us focus on a single entry $j$ in these three vectors, for which $s_j > 0$, and omit this index. We denote these entries as $\eta_{T,i}$, $n_{T,i}$ and $z_T$, respectively. We construct $\eta_{T,i}$ such that

$$\mathbb{E}\left[\eta_{T,i} \cdot z_T\right] = \begin{cases} \mathbb{E}\left[\eta_{T,i}^2\right] & \text{for } i \geq i_j \\ \mathbb{E}\left[\left(z_T - n_{T,i_j}\right)^2\right] & \text{for } i = i_j - 1 \\ 0 & \text{otherwise.} \end{cases}$$

This implies that the layers of noise $\eta_{T,L+1}, \ldots \eta_{T,i_j}$ are all portions of $z_T$ itself, with an additional portion being contained in $\eta_{T,i_j-1}$. Afterwards, $\eta_{T,i}$ become independent of $z_T$. In the case of $s_j = 0$, the above relations simplify to be $E[\eta_{T,i} \cdot z_T] = 0$ for all $i$, implying no statistical dependency between the given and the synthetic noises. Consequently, it can be shown that the overall noise in Equation 5 satisfies

$$\left(\boldsymbol{\Sigma}\mathbf{V}^T\mathbf{n}_i - \mathbf{z_T}\right)_j = n_{T,i} - z_T \sim \begin{cases} \mathcal{N}\left(0, s_j^2\sigma_i^2 - \sigma_0^2\right) & \text{if } \sigma_i s_j > \sigma_0 \\ \mathcal{N}\left(0, \sigma_0^2 - s_j^2\sigma_i^2\right) & \text{otherwise.} \end{cases} \tag{6}$$

The top option refers to high values of the annealed Langevin noise, in which, despite the possible decay caused by the singular value $s_j$, this noise is stronger than $z_T$. In this case, $n_{T,i}$ contains all $z_T$ and an additional independent portion of noise. The bottom part assumes that the annealed noise (with the influence of $s_j$) is weaker than the measurements' noise, and then it is fully immersed within $z_T$, with the difference being Gaussian and independent.

## 3.2 Derivation of the Conditional Score Function

The above derivations show that the noise in Equation 5 is zero-mean, Gaussian with uncorrelated entries and of known variance, and this noise is independent of $\tilde{\mathbf{x}}_i$. Thus Equation 5 can be used conveniently for deriving the measurements part of the conditional score function. We denote $\tilde{\mathbf{x}}_T = \mathbf{V}^T\tilde{\mathbf{x}}_i$, $\tilde{\mathbf{x}} = \tilde{\mathbf{x}}_i$, $\mathbf{n} = \mathbf{n}_i$ for simplicity, and turn to calculate $\nabla_{\tilde{\mathbf{x}}_T} \log p\left(\tilde{\mathbf{x}}_T|\mathbf{y}_T\right)$. We split $\tilde{\mathbf{x}}_T$ into three parts: (i) $\tilde{\mathbf{x}}_{T,0}$ refers to the entries $j$ for which $s_j = 0$; (ii) $\tilde{\mathbf{x}}_{T,<}$ corresponds to the entries $j$ for which $0 < \sigma_i s_j < \sigma_0$; and (iii) $\tilde{\mathbf{x}}_{T,>}$ includes the entries $j$ for which $\sigma_i s_j > \sigma_0$. Observe that this partition of the entries of $\tilde{\mathbf{x}}_T$ is non-overlapping and fully covering. Similarly, we partition every vector $\mathbf{v} \in \mathbb{R}^N$ into $\mathbf{v}_0, \mathbf{v}_<, \mathbf{v}_>$, which are the entries of $\mathbf{v}$ corresponding to $\tilde{\mathbf{x}}_{T,0}, \tilde{\mathbf{x}}_{T,<}, \tilde{\mathbf{x}}_{T,>}$, respectively. Furthermore, we define $\mathbf{v}_{\cancel{0}}, \mathbf{v}_{\cancel{<}}, \mathbf{v}_{\cancel{>}}$ as all the entries of $\mathbf{v}$ except $\mathbf{v}_0, \mathbf{v}_<, \mathbf{v}_>$, respectively. With these definitions in place, the complete derivation of the score function is detailed in the supplemental material, and here we bring the final outcome. For $\tilde{\mathbf{x}}_{T,0}$, the score is independent of the measurements and given by

$$\nabla_{\tilde{\mathbf{x}}_{T,0}} \log p\left(\tilde{\mathbf{x}}_T|\mathbf{y}_T\right) = \left(\mathbf{V}^T\nabla_{\tilde{\mathbf{x}}} \log p\left(\tilde{\mathbf{x}}\right)\right)_0. \tag{7}$$

For the case of $\tilde{\mathbf{x}}_{T,>}$, the expression obtained is only measurements-dependent,

$$\nabla_{\tilde{\mathbf{x}}_{T,>}} \log p\left(\tilde{\mathbf{x}}_T|\mathbf{y}_T\right) = \left(\boldsymbol{\Sigma}^T\left(\sigma_i^2\boldsymbol{\Sigma}\boldsymbol{\Sigma}^T - \sigma_0^2\mathbf{I}\right)^\dagger\left(\mathbf{y}_T - \boldsymbol{\Sigma}\tilde{\mathbf{x}}_T\right)\right)_>. \tag{8}$$

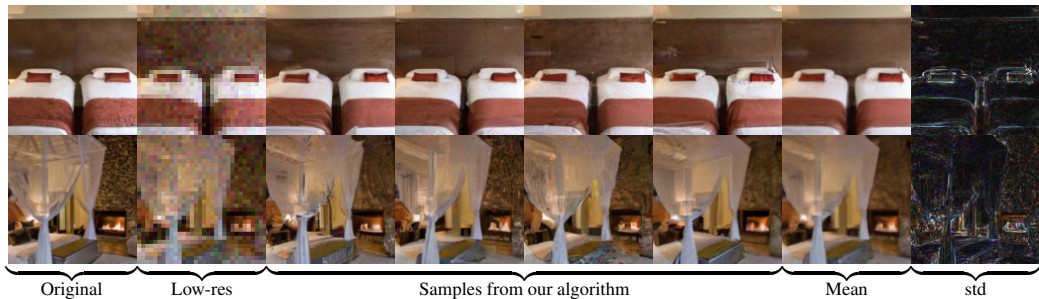

| Original | Low-res | Samples from our algorithm | Mean | std |

Figure 2: Super resolution results on LSUN bedroom [56] images (downscaling $4:1$ by plain averaging and adding noise with $\sigma_0 = 0.04$).

Lastly, for the case of $\tilde{\mathbf{x}}_{T,<}$, the conditional score includes two terms – one referring to the plain (blurred) score, and the other depending on the measurements,

$$\nabla_{\tilde{\mathbf{x}}_{T,<}} \log p\left(\tilde{\mathbf{x}}_T | \mathbf{y}_T\right) = \left(\mathbf{\Sigma}^T \left(\sigma_0^2 \mathbf{I} - \sigma_i^2 \mathbf{\Sigma}\mathbf{\Sigma}^T\right)^\dagger \left(\mathbf{y}_T - \mathbf{\Sigma}\tilde{\mathbf{x}}_T\right)\right)_< + \left(\mathbf{V}^T \nabla_{\tilde{\mathbf{x}}} \log p\left(\tilde{\mathbf{x}}\right)\right)_<. \quad (9)$$

As already mentioned, the full derivations of equations 7, 8, and 9 are detailed in the supplemental material. Aggregating all these results together, we obtain the following conditional score function:

$$\nabla_{\tilde{\mathbf{x}}_T} \log p\left(\tilde{\mathbf{x}}_T | \mathbf{y}_T\right) = \mathbf{\Sigma}^T \left|\sigma_0^2 \mathbf{I} - \sigma_i^2 \mathbf{\Sigma}\mathbf{\Sigma}^T\right|^\dagger \left(\mathbf{y}_T - \mathbf{\Sigma}\tilde{\mathbf{x}}_T\right) + \left(\mathbf{V}^T \nabla_{\tilde{\mathbf{x}}} \log p\left(\tilde{\mathbf{x}}\right)\right)\Big|_{\ngtr}, \quad (10)$$

where $(\mathbf{v})|_{\ngtr}$ is the vector $\mathbf{v}$, but with zeros in its entries that correspond to $\mathbf{v}_>$. Observe that the first term in Equation 10 contains zeros in the entries corresponding to $\tilde{\mathbf{x}}_{T,0}$, matching the above calculations. The vector $\nabla_{\tilde{\mathbf{x}}} \log p\left(\tilde{\mathbf{x}}\right)$ can be estimated using a neural network as in [44], or using a pre-trained MMSE denoiser as in [20, 21]. All the other elements of this vector are given or can be easily obtained from $\mathbf{H}$ by calculating its SVD decomposition once at the beginning.

## 4 The Proposed Algorithm

Armed with the conditional score function in Equation 10, the Langevin dynamics algorithm can be run with a constant step size or an annealed step size as in [44], and this should converge to a sample from $p\left(\tilde{\mathbf{x}}_T | \mathbf{y}_T\right)$. However, for this to perform well, one should use a very small step size, implying a devastatingly slow convergence behavior. This is mainly due to the fact that different entries of $\tilde{\mathbf{x}}_T$ advance at different speeds, in accord with their corresponding singular values. As the added noise in each step has the same variance in every entry, this leads an unbalanced signal-to-noise ratio, which considerably slows down the algorithm.

In order to mitigate this problem, we suggest using a *step size vector* $\boldsymbol{\alpha}_i \in \mathbb{R}^N$. We denote $\mathbf{A}_i = diag\left(\boldsymbol{\alpha}_i\right)$, and obtain the following update formula for a Langevin dynamics algorithm:

$$\mathbf{V}^T \tilde{\mathbf{x}}_i = \mathbf{V}^T \tilde{\mathbf{x}}_{i-1} + c \cdot \mathbf{A}_i \cdot \nabla_{\mathbf{V}^T \tilde{\mathbf{x}}_i} \log p\left(\mathbf{V}^T \tilde{\mathbf{x}}_i | \mathbf{y}_T\right) + \sqrt{2 \cdot c} \mathbf{A}_i^{\frac{1}{2}} \cdot \mathbf{z}_i, \quad (11)$$

where the conditional score function is estimated as described in subsection 3.2, and $c$ is some constant. For the choice of the step sizes in the diagonal of $\mathbf{A}_i$, we draw inspiration from Newton's method in optimization, which is designed to speed up convergence to local maximum points. The update formula in Newton's method is the same as Equation 11, but without the additional noise $\mathbf{z}_i$, and with $\mathbf{A}_i$ being the negative inverse Hessian of $\log p\left(\mathbf{V}^T \tilde{\mathbf{x}}_i | \mathbf{y}_T\right)$. We calculate a diagonal approximation of the Hessian, and set $\mathbf{A}_i$ to be its negative inverse. We also estimate the conditional score function using Equation 10 and a neural network. Note that this mixture of Langevin dynamics and Newton's method has been suggested in a slightly different context in [43], where the Hessian was approximated using a Quasi-Newton method. In our case, we analytically calculate a diagonal approximation of the negative inverse Hessian and obtain the following:

$$(\boldsymbol{\alpha}_i)_j = \begin{cases} \sigma_i^2, & s_j = 0 \\ \sigma_i^2 - \frac{\sigma_0^2}{s_j^2}, & \sigma_i s_j > \sigma_0 \\ \sigma_i^2 \cdot \left(1 - s_j^2 \frac{\sigma_i^2}{\sigma_0^2}\right), & 0 < \sigma_i s_j < \sigma_0. \end{cases} \quad (12)$$

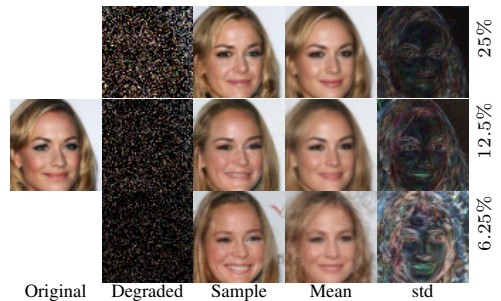

Original  Degraded  Sample  Mean  std

Figure 3: Compressive sensing results on a CelebA [27] image with an additive noise of $\sigma_0 = 0.1$.

The full derivations for each of the three cases are detailed in the supplemental material. Using these step sizes, the update formula in Equation 11, the conditional score function in Equation 10, and a neural network $\mathbf{s}(\tilde{\mathbf{x}}, \sigma)$ that estimates the score function $\nabla_{\tilde{\mathbf{x}}} \log p(\tilde{\mathbf{x}})$,[3] we obtain a tractable iterative algorithm for sampling from $p(\tilde{\mathbf{x}}_L \mid \mathbf{y})$, where the noise in $\tilde{\mathbf{x}}_L$ is sufficiently negligible to be considered as a sampling from the ideal image manifold.

---

**Algorithm 1:** SNIPS

---

**Input:** $\{\sigma_i\}_{i=1}^L$, $c$, $\tau$, $\mathbf{y}$, $\mathbf{H}$, $\sigma_0$

1  $\mathbf{U}, \boldsymbol{\Sigma}, \mathbf{V} \leftarrow svd(\mathbf{H})$
2  Initialize $\mathbf{x_0}$ with random noise $U[0,1]$
3  **for** $i \leftarrow 1$ *to* $L$ **do**
4  $\quad (\mathbf{A}_i)_0 \leftarrow \sigma_i^2 \mathbf{I}$
5  $\quad (\mathbf{A}_i)_< \leftarrow \sigma_i^2 \cdot \left(\mathbf{I} - \frac{\sigma_i^2}{\sigma_0^2} \boldsymbol{\Sigma}_< \boldsymbol{\Sigma}_<^T\right)$
6  $\quad (\mathbf{A}_i)_> \leftarrow \sigma_i^2 \mathbf{I} - \sigma_0^2 \boldsymbol{\Sigma}_>^\dagger \boldsymbol{\Sigma}_>^{\dagger^T}$
7  $\quad$ **for** $t \leftarrow 1$ *to* $\tau$ **do**
8  $\quad\quad$ Draw $\mathbf{z}_t \sim \mathcal{N}(0, \mathbf{I})$
9  $\quad\quad \mathbf{d}_t \leftarrow \boldsymbol{\Sigma}^T \cdot \left|\sigma_0^2 \mathbf{I} - \sigma_i^2 \boldsymbol{\Sigma}\boldsymbol{\Sigma}^T\right|^\dagger \cdot \left(\mathbf{U}^T\mathbf{y} - \boldsymbol{\Sigma}\mathbf{V}^T\mathbf{x}_{t-1}\right) + \left(\mathbf{V}^T \cdot \mathbf{s}(\mathbf{x}_{t-1}, \sigma_i)\right)\big|_{\not>}$
10  $\quad\quad \mathbf{x}_t \leftarrow \mathbf{V} \cdot \left(\mathbf{V}^T\mathbf{x}_{t-1} + c\mathbf{A}_i\mathbf{d}_t + \sqrt{2c}\mathbf{A}_i^{\frac{1}{2}}\mathbf{z}_t\right)$
11  $\quad$ **end**
12  $\quad \mathbf{x}_0 \leftarrow \mathbf{x}_\tau$
13  **end**

**Output:** $\mathbf{x_0}$

---

Note that when we set $\mathbf{H} = 0$ and $\sigma_0 = 0$, implying no measurements, the above algorithm degenerates to an image synthesis, exactly as in [44]. Two other special cases of this algorithm are obtained for $\mathbf{H} = \mathbf{I}$ or $\mathbf{H} = \mathbf{I}$ with some rows removed, the first referring to denoising and the second to noisy inpainting, both cases shown in [21]. Lastly, for the choices of $\mathbf{H}$ as in [20] or [44, 46] and with $\sigma_0 = 0$, the above algorithm collapses to a close variant of their proposed iterative methods.

## 5  Experimental Results

In our experiments we use the NCSNv2 [45] network in order to estimate the score function of the prior distribution. Three different NCSNv2 models are used, each trained separately on the training sets of: (i) images of size $64 \times 64$ pixels from the CelebA dataset [27]; (ii) images of size $128 \times 128$ pixels from LSUN [56] bedrooms dataset; and (iii) LSUN $128 \times 128$ images of towers. We demonstrate SNIPS' capabilities on the respective test sets for image deblurring, super resolution, and compressive sensing. In each of the experiments, we run our algorithm 8 times, producing 8 samples for each input. We examine both the samples themselves and their mean, which serves as an approximation of the MMSE solution, $\mathbb{E}[\mathbf{x}|\mathbf{y}]$.

---

[3]Recall that $\mathbf{s}(\tilde{\mathbf{x}}, \sigma) = (\mathbf{D}(\tilde{\mathbf{x}}, \sigma) - \tilde{\mathbf{x}})/\sigma^2$, being a denoising residual.

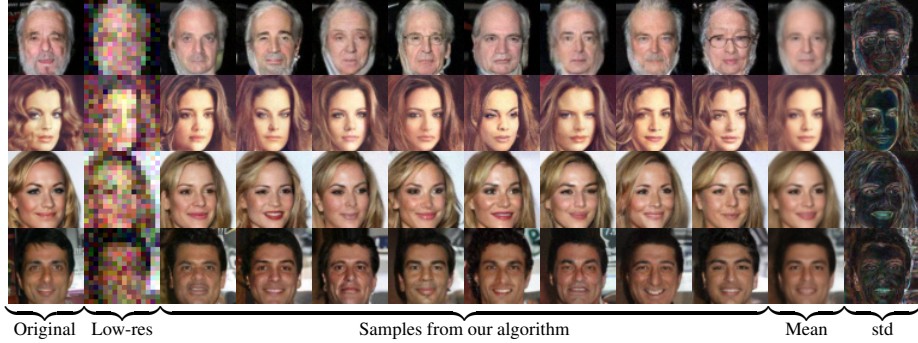

Original Low-res        Samples from our algorithm      Mean std

Figure 4: Super resolution results on CelebA [27] images (downscaling $4:1$ by plain averaging and adding noise with $\sigma_0 = 0.1$).

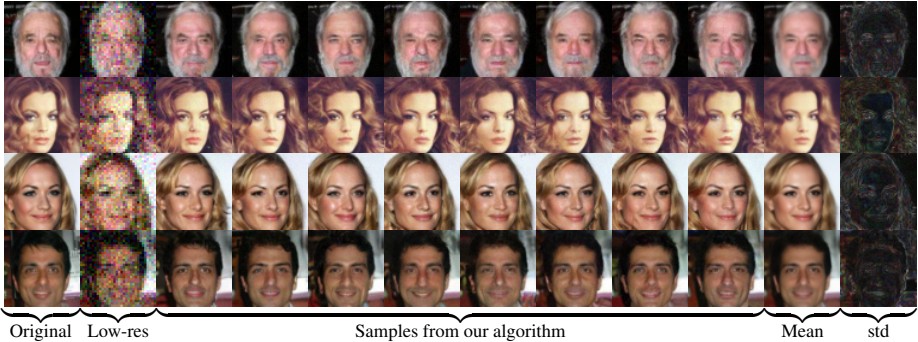

Original Low-res        Samples from our algorithm      Mean std

Figure 5: Super resolution results on CelebA [27] images (downscaling $2:1$ by plain averaging and adding noise with $\sigma_0 = 0.1$).

**For image deblurring**, we use a uniform $5 \times 5$ blur kernel, and an additive white Gaussian noise with $\sigma_0 = 0.1$ (referring to pixel values in the range $[0, 1]$). Figure 1 demonstrates the obtained results for several images taken from the CelebA dataset. As can be seen, SNIPS produces visually pleasing, diverse samples.

**For super resolution**, the images are downscaled using a block averaging filter, *i.e.*, each non-overlapping block of pixels in the original image is averaged into one pixel in the low-resolution image. We use blocks of size $2 \times 2$ or $4 \times 4$ pixels, and assume the low-resolution image to include an additive white Gaussian noise. We showcase results on LSUN and CelebA in Figures 2, 4, and 5.

**For compressive sensing**, we use three random projection matrices with singular values of $1$, that compress the image by $25\%$, $12.5\%$, and $6.25\%$. As can be seen in Figure 3 and as expected, the more aggressive the compression, the more significant are the variations in reconstruction.

We calculate the average PSNR (peak signal-to-noise ratio) of each of the $8$ samples in our experiments, as well as the PSNR of their mean, as shown in Table 1. In all the experiments, the empirical conditional mean presents an improvement of around $2.4$ dB in PSNR, even though it is less visually appealing compared to the samples. This is consistent with the theory in [5], which states that the difference in PSNR between posterior samples and the conditional mean (the MMSE estimator) should be 3 dB, with the MMSE estimator having poorer perceptual quality but better PSNR.

A comparison of our deblurring results to those obtained by RED [39] is detailed in the supplemental material. We show that SNIPS exhibits superior performance over RED, achieving more than $11\%$ improvement in PSNR and more than $58\%$ improvement in LPIPS [62], a perceptual quality metric.

### 5.1 Assessing Faithfulness to the Measurements

A valid solution to an inverse problem should satisfy two conditions: (i) It should be visually pleasing, consistent with the underlying prior distribution of images, and (ii) It should be faithful to the given measurement, maintaining the relationship as given in the problem setting. Since the prior distribution

Table 1: PSNR results for different inverse problems on 8 images from CelebA [27]. We ran SNIPS 8 times, and obtained 8 samples. The average PSNR for each of the samples is in the first column, while the average PSNR for the mean of the 8 samples for each image is in the second one.

| Problem | Sample PSNR | Mean PSNR |
|---|---|---|
| Uniform deblurring | 25.54 | 28.01 |
| Super resolution (by 2) | 25.58 | 28.03 |
| Super resolution (by 4) | 21.90 | 24.31 |
| Compressive sensing (by 25%) | 25.68 | 28.06 |
| Compressive sensing (by 12.5%) | 22.34 | 24.67 |

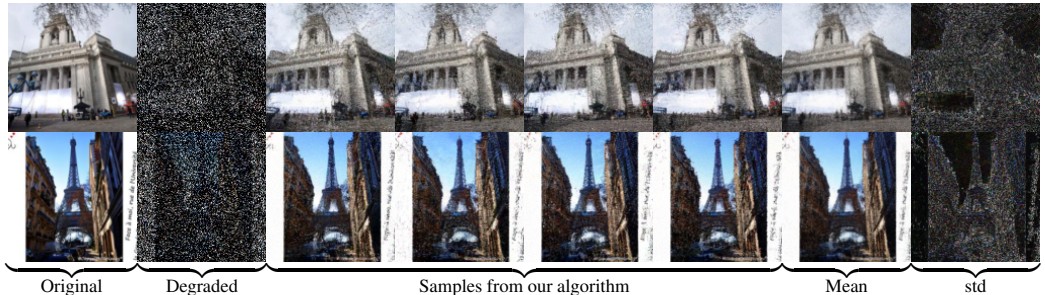

Figure 6: Compressive sensing results on LSUN [56] tower images (compression by $25\%$ and adding noise with $\sigma_0 = 0.04$).

is unknown, we assess the first condition by visually observing the obtained solutions and their tendency to look realistic. As for the second condition, we perform the following computation: We degrade the obtained reconstruction $\hat{x}$ by $\mathbf{H}$, and calculate its difference from the given measurement $\mathbf{y}$, obtaining $\mathbf{y} - \mathbf{H}\hat{x}$. According to the problem setting, this difference should be an additive white Gaussian noise vector with a standard deviation of $\sigma_0$. We examine this difference by calculating its empirical standard deviation, and performing the Pearson-D'Agostino [8] test of normality on it, accepting it as a Gaussian vector if the obtained p-value is greater than $0.05$. We also calculate the Pearson correlation coefficient (denoted as $\rho$) among neighboring entries, accepting them as uncorrelated for coefficients smaller than $0.1$ in absolute value. In all of our tests, the standard deviation matches $\sigma_0$ almost exactly, the Pearson correlation coefficient satisfies $|\rho| < 0.1$, and we obtain p-values greater than $0.05$ in around $95\%$ of the samples (across all experiments). These results empirically show that our algorithm produces valid solutions to the given inverse problems.

## 6   Conclusion and Future Work

SNIPS, presented in this paper, is a novel stochastic algorithm for solving general noisy linear inverse problems. This method is based on annealed Langevin dynamics and Newton's method, and relies on the availability of a pre-trained Gaussian MMSE denoiser. SNIPS produces a random variety of high quality samples from the posterior distribution of the unknown given the measurements, while guaranteeing their validity with respect to the given data. This algorithm's derivation includes an intricate choice of the injected annealed noise in the Langevin update equations, and an SVD decomposition of the degradation operator for decoupling the measurements' dependencies. We demonstrate SNIPS' success on image deblurring, super resolution, and compressive sensing.

Extensions of this work should focus on SNIPS' limitations: (i) The need to deploy SVD decomposition of the degradation matrix requires a considerable amount of memory and computations, and hinders the algorithm's scalability; (ii) The current version of SNIPS does not handle general content images, a fact that is related to the properties of the denoiser being used [41]; and (iii) SNIPS, as any other Langevin based method, requires (too) many iterations (*e.g.*, in our super-resolution tests on CelebA, 2 minutes are required for producing 8 sample images), and means for its acceleration should be explored.

## 7 Funding Transparency Statement

This research was partially supported by the Israel Science Foundation (ISF) under Grant 335/18 and the Technion Hiroshi Fujiwara Cyber Security Research Center and the Israel Cyber Bureau. Bahjat Kawar's scholarship was partially provided by Li Ka Shing Fellowships and the Planning and Budgeting Committee of the Israel Council for Higher Education.

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
