# SNIPS: Solving Noisy Inverse Problems Stochastically (Appendices)

**Bahjat Kawar, Gregory Vaksman, Michael Elad**
Computer Science Department, Technion, Haifa, Israel
{bahjat.kawar, grishav, elad}@cs.technion.ac.il

## A  Conditional Score Derivation Proofs

We would like to derive a term for $\nabla \tilde{\mathbf{x}}_T \log p\left(\tilde{\mathbf{x}}_T | \mathbf{y}_T\right)$ depending on known ingredients such as $\tilde{\mathbf{x}}_T$, $\mathbf{y}_T$, $\sigma_0$, $\sigma_i$ and the SVD components of $\mathbf{H}$, as well as the blurred prior score function $\nabla \tilde{\mathbf{x}}_T \log p\left(\tilde{\mathbf{x}}_T\right)$, which can be estimated using a neural network. To that end, in accordance with the definitions of $\mathbf{v}_0$, $\mathbf{v}_<$ and $\mathbf{v}_>$, for a matrix $\mathbf{M}$ we define $\mathbf{M}_0, \mathbf{M}_<, \mathbf{M}_>$ as leading minors of $\mathbf{M}$ with subsets of rows and columns extracted accordingly from the above-defined partition. Recalling Equation 5 from the main paper, we have

$$\mathbf{y}_T - \boldsymbol{\Sigma}\tilde{\mathbf{x}}_T = \mathbf{U}^T\mathbf{z} - \boldsymbol{\Sigma}\mathbf{V}^T\mathbf{n}. \tag{1}$$

Observe that the entries of the right-hand-side vector are statistically independent, and their distribution for $s_j < \sigma_0/\sigma_i$ is given by

$$\left(\mathbf{U}^T\mathbf{z} - \boldsymbol{\Sigma}\mathbf{V}^T\mathbf{n}\right)_< \sim \mathcal{N}\left(0, \sigma_0^2\mathbf{I} - \sigma_i^2\boldsymbol{\Sigma}_<\boldsymbol{\Sigma}_<^T\right). \tag{2}$$

This is a direct result of Equation 6 in the main paper, obtained by simply aggregating the different entries $j$ into a vector. Similarly,

$$\left(\mathbf{V}^T\mathbf{n} - \boldsymbol{\Sigma}^\dagger\mathbf{U}^T\mathbf{z}\right)_> \sim \mathcal{N}\left(0, \sigma_i^2\mathbf{I} - \sigma_0^2\boldsymbol{\Sigma}_>^{-1}\boldsymbol{\Sigma}_>^{-1^T}\right), \tag{3}$$

obtained from aggregating the entries from Equation 6 in the main paper into a vector, and multiplying it by $\boldsymbol{\Sigma}_>^{-1}$. Notice that $\boldsymbol{\Sigma}_>$ is a diagonal square matrix, and thus invertible. The above two formulae will be used in the following analysis.

**Theorem 1.** *Given* $\mathbf{y} = \mathbf{H}\mathbf{x} + \mathbf{z}$, $\mathbf{z} \sim \mathcal{N}\left(0, \sigma_0^2\mathbf{I}\right)$, $\mathbf{H} = \mathbf{U}\boldsymbol{\Sigma}\mathbf{V}^T$ *is the SVD decomposition of* $\mathbf{H}$, $\mathbf{y}_T = \mathbf{U}^T\mathbf{y}$, $\mathbf{n} = \mathbf{n}_i$ *as constructed in subsection 3.1 in the main paper,* $\tilde{\mathbf{x}} = \tilde{\mathbf{x}}_i = \mathbf{x} + \mathbf{n}$, $\tilde{\mathbf{x}}_T = \mathbf{V}^T\tilde{\mathbf{x}}$, $\mathbf{x}_T = \mathbf{V}^T\mathbf{x}$, *the conditional score is approximately given by:*

$$\nabla_{\tilde{\mathbf{x}}_T} \log p\left(\tilde{\mathbf{x}}_T | \mathbf{y}_T\right) = \boldsymbol{\Sigma}^T\left|\sigma_0^2\mathbf{I} - \sigma_i^2\boldsymbol{\Sigma}\boldsymbol{\Sigma}^T\right|^\dagger\left(\mathbf{y}_T - \boldsymbol{\Sigma}\tilde{\mathbf{x}}_T\right) + \left.\left(\mathbf{V}^T\nabla_{\tilde{\mathbf{x}}} \log p\left(\tilde{\mathbf{x}}\right)\right)\right|_{\not>}$$

*Proof.* We split our derivation into three cases: $\tilde{\mathbf{x}}_{T,0}$, $\tilde{\mathbf{x}}_{T,>}$, and $\tilde{\mathbf{x}}_{T,<}$, and then concatenate the results.

**For the case of** $\tilde{\mathbf{x}}_{T,0}$, we calculate using the Bayes rule:

$$\nabla_{\tilde{\mathbf{x}}_{T,0}} \log p\left(\tilde{\mathbf{x}}_T | \mathbf{y}_T\right) = \nabla_{\tilde{\mathbf{x}}_{T,0}} \log p\left(\mathbf{y}_T | \tilde{\mathbf{x}}_T\right) + \nabla_{\tilde{\mathbf{x}}_{T,0}} \log p\left(\tilde{\mathbf{x}}_T\right).$$

Deriving by $\tilde{\mathbf{x}}_{T,0}$ is the same as deriving by $\tilde{\mathbf{x}}_T$ and then taking the part referring to zero singular values of $\mathbf{H}$. Thus, the second term becomes $\left(\nabla_{\tilde{\mathbf{x}}_T} \log p\left(\tilde{\mathbf{x}}_T\right)\right)_0$. As for the first term, we can subtract the vector $\boldsymbol{\Sigma}\tilde{\mathbf{x}}_T$ without changing the statistics because it is a known quantity in this setting, resulting in

$$\begin{aligned}\nabla_{\tilde{\mathbf{x}}_{T,0}} \log p\left(\tilde{\mathbf{x}}_T | \mathbf{y}_T\right) &= \nabla_{\tilde{\mathbf{x}}_{T,0}} \log p\left(\mathbf{y}_T - \boldsymbol{\Sigma}\tilde{\mathbf{x}}_T | \tilde{\mathbf{x}}_T\right) + \left(\nabla_{\tilde{\mathbf{x}}_T} \log p\left(\tilde{\mathbf{x}}_T\right)\right)_0 \\ &= \nabla_{\tilde{\mathbf{x}}_{T,0}} \log p\left(\mathbf{U}^T\mathbf{z} - \boldsymbol{\Sigma}\mathbf{V}^T\mathbf{n} | \tilde{\mathbf{x}}_T\right) + \left(\nabla_{\tilde{\mathbf{x}}_T} \log p\left(\tilde{\mathbf{x}}_T\right)\right)_0.\end{aligned}$$

35th Conference on Neural Information Processing Systems (NeurIPS 2021).

The last equality holds due to Equation 1. Referring to the first term, because the entries of the vector are independent, we can split the probability density function into a product of two such functions for two parts of the vector, as follows:

$$\nabla_{\tilde{\mathbf{x}}_{T,0}} \left[ \log p \left( \left( \mathbf{U}^T \mathbf{z} - \boldsymbol{\Sigma} \mathbf{V}^T \mathbf{n} \right)_0 | \tilde{\mathbf{x}}_T \right) + \log p \left( \left( \mathbf{U}^T \mathbf{z} - \boldsymbol{\Sigma} \mathbf{V}^T \mathbf{n} \right)_{0\!\!\!/} | \tilde{\mathbf{x}}_T \right) \right].$$

The entries of $\left( \mathbf{U}^T \mathbf{z} - \boldsymbol{\Sigma} \mathbf{V}^T \mathbf{n} \right)_{0\!\!\!/}$ were defined element-wise as gradual noise additions, statistically independent of the entries of $\tilde{\mathbf{x}}_{T,0}$. Therefore, the conditioning on $\tilde{\mathbf{x}}_T$ is equivalent to conditioning on $\tilde{\mathbf{x}}_{T,0\!\!\!/}$. Deriving this log-probability by $\tilde{\mathbf{x}}_{T,0}$ results in zero. As for the first term, $\left( \boldsymbol{\Sigma} \mathbf{V}^T \mathbf{n} \right)_0$ is zero due to the definition of $\boldsymbol{\Sigma}$, and $\left( \mathbf{U}^T \mathbf{z} \right)_0$ is a Gaussian vector that is independent of $\tilde{\mathbf{x}}_{T,0}$, resulting in

$$\begin{aligned}
\nabla_{\tilde{\mathbf{x}}_{T,0}} \log p \left( \tilde{\mathbf{x}}_T | \mathbf{y}_T \right) &= \nabla_{\tilde{\mathbf{x}}_{T,0}} \log p \left( \left( \mathbf{U}^T \mathbf{z} - \boldsymbol{\Sigma} \mathbf{V}^T \mathbf{n} \right)_0 | \tilde{\mathbf{x}}_T \right) + \left( \nabla_{\tilde{\mathbf{x}}_T} \log p \left( \tilde{\mathbf{x}}_T \right) \right)_0 \\
&= \nabla_{\tilde{\mathbf{x}}_{T,0}} \log p \left( \left( \mathbf{U}^T \mathbf{z} \right)_0 | \tilde{\mathbf{x}}_T \right) + \left( \nabla_{\tilde{\mathbf{x}}_T} \log p \left( \tilde{\mathbf{x}}_T \right) \right)_0 \\
&= \left( \nabla_{\tilde{\mathbf{x}}_T} \log p \left( \tilde{\mathbf{x}}_T \right) \right)_0 \\
&= \left( \nabla_{\tilde{\mathbf{x}}_T} \log p \left( \tilde{\mathbf{x}} \right) \right)_0 \\
&= \left( \mathbf{V}^T \nabla_{\tilde{\mathbf{x}}} \log p \left( \tilde{\mathbf{x}} \right) \right)_0 .
\end{aligned}$$

The second last equality holds because $\tilde{\mathbf{x}} = \mathbf{V} \tilde{\mathbf{x}}_T$, and multiplication by the orthogonal matrix $\mathbf{V}$ does not change the statistics of the variable. The last equality holds due to the multivariate chain rule: $\nabla_{\mathbf{x}} f \left( \mathbf{y} \right) = \mathbf{J} \left( \mathbf{y} \left( \mathbf{x} \right) \right) \nabla_{\mathbf{y}} f \left( \mathbf{y} \right)$, where $\mathbf{J} \left( \mathbf{y} \left( \mathbf{x} \right) \right)$ is the Jacobian matrix of $\mathbf{y}$ w.r.t. $\mathbf{x}$. Finally, we obtain

$$\nabla_{\tilde{\mathbf{x}}_{T,0}} \log p \left( \tilde{\mathbf{x}}_T | \mathbf{y}_T \right) = \left( \mathbf{V}^T \nabla_{\tilde{\mathbf{x}}} \log p \left( \tilde{\mathbf{x}} \right) \right)_0 . \tag{4}$$

**For the case of $\tilde{\mathbf{x}}_{T,>}$**, using the definition of the conditional distribution we get:

$$\begin{aligned}
\nabla_{\tilde{\mathbf{x}}_{T,>}} \log p \left( \tilde{\mathbf{x}}_T | \mathbf{y}_T \right) &= \nabla_{\tilde{\mathbf{x}}_{T,>}} \log p \left( \tilde{\mathbf{x}}_{T,0}, \tilde{\mathbf{x}}_{T,0\!\!\!/} | \mathbf{y}_T \right) \\
&= \nabla_{\tilde{\mathbf{x}}_{T,>}} \log p \left( \tilde{\mathbf{x}}_{T,0} | \tilde{\mathbf{x}}_{T,0\!\!\!/}, \mathbf{y}_T \right) + \nabla_{\tilde{\mathbf{x}}_{T,>}} \log p \left( \tilde{\mathbf{x}}_{T,0\!\!\!/} | \mathbf{y}_T \right) .
\end{aligned} \tag{5}$$

Focusing on the second term, we calculate, with a similar reasoning as above and get:

$$\nabla_{\tilde{\mathbf{x}}_{T,>}} \log p \left( \tilde{\mathbf{x}}_{T,0\!\!\!/} | \mathbf{y}_T \right) = \nabla_{\tilde{\mathbf{x}}_{T,>}} \log p \left( \left( \tilde{\mathbf{x}}_T - \boldsymbol{\Sigma}^\dagger \mathbf{y}_T \right)_{0\!\!\!/} | \mathbf{y}_T \right) .$$

Substituting $\tilde{\mathbf{x}}_T = \mathbf{V}^T \mathbf{x} + \mathbf{V}^T \mathbf{n}$, $\mathbf{y}_T = \mathbf{U}^T \mathbf{H} \mathbf{x} + \mathbf{U}^T \mathbf{z}$, $\mathbf{H} = \mathbf{U} \boldsymbol{\Sigma} \mathbf{V}^T$ leads to

$$\begin{aligned}
\nabla_{\tilde{\mathbf{x}}_{T,>}} \log p \left( \tilde{\mathbf{x}}_{T,0\!\!\!/} | \mathbf{y}_T \right) &= \nabla_{\tilde{\mathbf{x}}_{T,>}} \log p \left( \left( \mathbf{V}^T \mathbf{x} + \mathbf{V}^T \mathbf{n} - \boldsymbol{\Sigma}^\dagger \left( \mathbf{U}^T \mathbf{H} \mathbf{x} + \mathbf{U}^T \mathbf{z} \right) \right)_{0\!\!\!/} | \mathbf{y}_T \right) \\
&= \nabla_{\tilde{\mathbf{x}}_{T,>}} \log p \left( \left( \mathbf{V}^T \mathbf{x} + \mathbf{V}^T \mathbf{n} - \boldsymbol{\Sigma}^\dagger \mathbf{U}^T \mathbf{U} \boldsymbol{\Sigma} \mathbf{V}^T \mathbf{x} - \boldsymbol{\Sigma}^\dagger \mathbf{U}^T \mathbf{z} \right)_{0\!\!\!/} | \mathbf{y}_T \right) \\
&= \nabla_{\tilde{\mathbf{x}}_{T,>}} \log p \left( \left( \mathbf{V}^T \mathbf{n} - \boldsymbol{\Sigma}^\dagger \mathbf{U}^T \mathbf{z} + \left( \mathbf{I} - \boldsymbol{\Sigma}^\dagger \boldsymbol{\Sigma} \right) \mathbf{x}_T \right)_{0\!\!\!/} | \mathbf{y}_T \right) .
\end{aligned}$$

The last equality holds because $\mathbf{U}^T \mathbf{U} = \mathbf{I}$. Observe that $\left( \mathbf{I} - \boldsymbol{\Sigma}^\dagger \boldsymbol{\Sigma} \right) \mathbf{x}_T$ is zero everywhere except in the $0$ part of the vector, which we discard because of the $0\!\!\!/$ notation. We can split this term into two parts, as before,

$$\nabla_{\tilde{\mathbf{x}}_{T,>}} \left[ \log p \left( \left( \mathbf{V}^T \mathbf{n} - \boldsymbol{\Sigma}^\dagger \mathbf{U}^T \mathbf{z} \right)_> | \mathbf{y}_T \right) + \log p \left( \left( \mathbf{V}^T \mathbf{n} - \boldsymbol{\Sigma}^\dagger \mathbf{U}^T \mathbf{z} \right)_< | \mathbf{y}_T \right) \right].$$

The derivative of the second term (the $<$ part) is zero, because this vector was built element-wise as gradual noise additions, independent of $\tilde{\mathbf{x}}_{T,>}$. This results in

$$\begin{aligned}
\nabla_{\tilde{\mathbf{x}}_{T,>}} \log p \left( \tilde{\mathbf{x}}_{T,0\!\!\!/} | \mathbf{y}_T \right) &= \nabla_{\tilde{\mathbf{x}}_{T,>}} \log p \left( \left( \mathbf{V}^T \mathbf{n} - \boldsymbol{\Sigma}^\dagger \mathbf{U}^T \mathbf{z} \right)_> | \mathbf{y}_T \right) \\
&= \nabla_{\tilde{\mathbf{x}}_{T,>}} \log p \left( \left( \tilde{\mathbf{x}}_T - \boldsymbol{\Sigma}^\dagger \mathbf{y}_T \right)_> | \mathbf{y}_T \right)
\end{aligned}$$

This is the gradient-log of a Gaussian density function of the vector $\left( \tilde{\mathbf{x}}_T - \boldsymbol{\Sigma}^\dagger \mathbf{y}_T \right)_>$, known to have a zero mean and a covariance matrix $\sigma_i^2 \mathbf{I} - \sigma_0^2 \boldsymbol{\Sigma}_>^{-1} \boldsymbol{\Sigma}_>^{-1^T}$, according to Equation 3. Thus, we use the

known Gaussian gradient-log and conclude:

$$\nabla_{\tilde{\mathbf{x}}_{T,>}} \log p\left(\tilde{\mathbf{x}}_{T,\not{0}}|\mathbf{y}_T\right) = \left(\sigma_i^2 \mathbf{I} - \sigma_0^2 \mathbf{\Sigma}_>^{-1} \mathbf{\Sigma}_>^{-1^T}\right)^{-1} \left(\mathbf{\Sigma}^\dagger \mathbf{y}_T - \tilde{\mathbf{x}}_T\right)_>$$

$$= \left(\mathbf{\Sigma}_>^{-1}\left(\mathbf{\Sigma}_>\sigma_i^2 \mathbf{I} \mathbf{\Sigma}_>^T - \sigma_0^2 \mathbf{I}\right)\mathbf{\Sigma}_>^{-1^T}\right)^{-1}\left(\mathbf{\Sigma}^\dagger \mathbf{y}_T - \tilde{\mathbf{x}}_T\right)_>$$

$$= \mathbf{\Sigma}_>^T \left(\sigma_i^2 \mathbf{\Sigma}_> \mathbf{\Sigma}_>^T - \sigma_0^2 \mathbf{I}\right)^{-1} \mathbf{\Sigma}_> \left(\mathbf{\Sigma}^\dagger \mathbf{y}_T - \tilde{\mathbf{x}}_T\right)_> .$$

Multiplying a certain part of a diagonal matrix (in this case, the $>$ part) by the corresponding part of a vector is the same as multiplying the original matrix and vector, and then taking the relevant part. This results in

$$\nabla_{\tilde{\mathbf{x}}_{T,>}} \log p\left(\tilde{\mathbf{x}}_{T,\not{0}}|\mathbf{y}_T\right) = \mathbf{\Sigma}_>^T \left(\sigma_i^2 \mathbf{\Sigma}_> \mathbf{\Sigma}_>^T - \sigma_0^2 \mathbf{I}\right)^{-1}\left(\mathbf{\Sigma}\mathbf{\Sigma}^\dagger \mathbf{y}_T - \mathbf{\Sigma}\tilde{\mathbf{x}}_T\right)_>$$

$$= \mathbf{\Sigma}_>^T \left(\sigma_i^2 \mathbf{\Sigma}_> \mathbf{\Sigma}_>^T - \sigma_0^2 \mathbf{I}\right)^{-1}\left(\mathbf{y}_T - \mathbf{\Sigma}\tilde{\mathbf{x}}_T\right)_> \qquad (6)$$

$$= \left(\mathbf{\Sigma}^T \left(\sigma_i^2 \mathbf{\Sigma}\mathbf{\Sigma}^T - \sigma_0^2 \mathbf{I}\right)^{-1}\left(\mathbf{y}_T - \mathbf{\Sigma}\tilde{\mathbf{x}}_T\right)\right)_> .$$

As for the first term in Equation 5, which is $\nabla_{\tilde{\mathbf{x}}_{T,>}} \log p\left(\tilde{\mathbf{x}}_{T,0}|\tilde{\mathbf{x}}_{T,\not{0}}, \mathbf{y}_T\right)$, we can rewrite it as $\nabla_{\tilde{\mathbf{x}}_{T,>}} \log p\left(\tilde{\mathbf{x}}_{T,0}|\tilde{\mathbf{x}}_{T,\not{0}}, \mathbf{\Sigma}_{\not{0}}^{-1}\mathbf{y}_T\right)$ because $\mathbf{\Sigma}_{\not{0}}^{-1}$ is an orthogonal matrix that does not add or remove information. Furthermore, we notice that the difference $\tilde{\mathbf{x}}_{T,\not{0}} - \mathbf{\Sigma}_{\not{0}}^{-1}\mathbf{y}_T$ was defined element-wise as gradual noise additions, independent of $\tilde{\mathbf{x}}_{T,0}$. Therefore, the term can be rewritten as $\nabla_{\tilde{\mathbf{x}}_{T,>}} \log p\left(\tilde{\mathbf{x}}_{T,0}|\tilde{\mathbf{x}}_{T,\not{0}}\right)$. We calculate using the definition of the conditional distribution:

$$\nabla_{\tilde{\mathbf{x}}_{T,>}} \log p\left(\tilde{\mathbf{x}}_{T,0}|\tilde{\mathbf{x}}_{T,\not{0}}\right) = \nabla_{\tilde{\mathbf{x}}_{T,>}} \log \frac{p\left(\tilde{\mathbf{x}}_{T,0}, \tilde{\mathbf{x}}_{T,\not{0}}\right)}{p\left(\tilde{\mathbf{x}}_{T,\not{0}}\right)}$$

$$= \nabla_{\tilde{\mathbf{x}}_{T,>}} \log p\left(\tilde{\mathbf{x}}_{T,0}, \tilde{\mathbf{x}}_{T,\not{0}}\right) - \nabla_{\tilde{\mathbf{x}}_{T,>}} \log p\left(\tilde{\mathbf{x}}_{T,\not{0}}\right)$$

$$= \nabla_{\tilde{\mathbf{x}}_{T,>}} \log p\left(\tilde{\mathbf{x}}_T\right) - \nabla_{\tilde{\mathbf{x}}_{T,>}} \log p\left(\tilde{\mathbf{x}}_{T,\not{0}}\right)$$

$$= \left(\nabla_{\tilde{\mathbf{x}}_T} \log p\left(\tilde{\mathbf{x}}_T\right)\right)_> - \left(\nabla_{\tilde{\mathbf{x}}_{T,\not{0}}} \log p\left(\tilde{\mathbf{x}}_{T,\not{0}}\right)\right)_>$$

$$= \left(\mathbf{V}^T \nabla_{\tilde{\mathbf{x}}} \log p\left(\tilde{\mathbf{x}}\right)\right)_> - \left(\mathbf{V}_{\not{0}}^T \nabla_{\tilde{\mathbf{x}}_{\not{0}}} \log p\left(\tilde{\mathbf{x}}_{\not{0}}\right)\right)_>$$

$$= \mathbf{V}_>^T \left(\nabla_{\tilde{\mathbf{x}}} \log p\left(\tilde{\mathbf{x}}\right)\right)_> - \mathbf{V}_>^T \left(\nabla_{\tilde{\mathbf{x}}_{\not{0}}} \log p\left(\tilde{\mathbf{x}}_{\not{0}}\right)\right)_> .$$

The second last equality holds due to the chain rule, and the last one holds because multiplying the $>$ part of a diagonal matrix by the corresponding part of a vector is the same as multiplying the original matrix and vector, and then taking the relevant part, as previously mentioned. Recalling Equation 3 in the main paper, we can substitute both terms by their denoiser counterparts, obtaining

$$\nabla_{\tilde{\mathbf{x}}_{T,>}} \log p\left(\tilde{\mathbf{x}}_{T,0}|\tilde{\mathbf{x}}_{T,\not{0}}\right) = \mathbf{V}_>^T \left(\frac{\mathbb{E}\left[\mathbf{x}|\tilde{\mathbf{x}}\right] - \tilde{\mathbf{x}}}{\sigma_i^2}\right)_> - \mathbf{V}_>^T \left(\frac{\mathbb{E}\left[\mathbf{x}_{\not{0}}|\tilde{\mathbf{x}}_{\not{0}}\right] - \tilde{\mathbf{x}}_{\not{0}}}{\sigma_i^2}\right)_>$$

$$= \frac{1}{\sigma_i^2} \mathbf{V}_>^T \left(\left(\mathbb{E}\left[\mathbf{x}|\tilde{\mathbf{x}}\right]\right)_> - \tilde{\mathbf{x}}_> - \left(\mathbb{E}\left[\mathbf{x}_{\not{0}}|\tilde{\mathbf{x}}_{\not{0}}\right]\right)_> + \tilde{\mathbf{x}}_>\right)$$

$$= \frac{1}{\sigma_i^2} \mathbf{V}_>^T \left(\left(\mathbb{E}\left[\mathbf{x}|\tilde{\mathbf{x}}\right]\right)_> - \left(\mathbb{E}\left[\mathbf{x}_{\not{0}}|\tilde{\mathbf{x}}_{\not{0}}\right]\right)_>\right)$$

$$= \frac{1}{\sigma_i^2} \mathbf{V}_>^T \left(\mathbb{E}\left[\mathbf{x}_>|\tilde{\mathbf{x}}\right] - \mathbb{E}\left[\mathbf{x}_>|\tilde{\mathbf{x}}_{\not{0}}\right]\right) .$$

We obtained a difference betwen two terms, both of which calculate an expectation of $\mathbf{x}_>$ given $\tilde{\mathbf{x}}_{\not{0}}$, with the first term including the extra knowledge of $\tilde{\mathbf{x}}_0$. We introduce an assumption that this additional information does not significantly change the estimation of $\mathbf{x}_>$, especially since a noisy version of it, $\tilde{\mathbf{x}}_>$, is given in both estimators. As a result, we obtain the approximation

$$\nabla_{\tilde{\mathbf{x}}_{T,>}} \log p\left(\tilde{\mathbf{x}}_{T,0}|\tilde{\mathbf{x}}_{T,\not{0}}\right) \approx \mathbf{0}. \qquad (7)$$

To conclude this part, we combine Equations 5, 6 and 7 and obtain the approximate relation

$$\nabla_{\tilde{\mathbf{x}}_{T,>}} \log p\left(\tilde{\mathbf{x}}_T|\mathbf{y}_T\right) = \left(\mathbf{\Sigma}^T \left(\sigma_i^2 \mathbf{\Sigma}\mathbf{\Sigma}^T - \sigma_0^2 \mathbf{I}\right)^{-1}\left(\mathbf{y}_T - \mathbf{\Sigma}\tilde{\mathbf{x}}_T\right)\right)_> . \qquad (8)$$

**For the case of $\tilde{\mathbf{x}}_{T,<}$**, we calculate using the Bayes rule, with similar reasoning to previous cases:

$$\nabla_{\tilde{\mathbf{x}}_{T,<}} \log p\left(\tilde{\mathbf{x}}_T | \mathbf{y}_T\right) = \nabla_{\tilde{\mathbf{x}}_{T,<}} \log p\left(\mathbf{y}_T | \tilde{\mathbf{x}}_T\right) + \nabla_{\tilde{\mathbf{x}}_{T,<}} \log p\left(\tilde{\mathbf{x}}_T\right)$$
$$= \nabla_{\tilde{\mathbf{x}}_{T,<}} \log p\left(\mathbf{y}_T - \mathbf{\Sigma}\tilde{\mathbf{x}}_T | \tilde{\mathbf{x}}_T\right) + \left(\nabla_{\tilde{\mathbf{x}}_T} \log p\left(\tilde{\mathbf{x}}_T\right)\right)_<$$
$$= \nabla_{\tilde{\mathbf{x}}_{T,<}} \log p\left(\mathbf{U}^T\mathbf{z} - \mathbf{\Sigma}\mathbf{V}^T\mathbf{n} | \tilde{\mathbf{x}}_T\right) + \left(\nabla_{\tilde{\mathbf{x}}_T} \log p\left(\tilde{\mathbf{x}}_T\right)\right)_< .$$

Similar to the first case, we can split the first term in the same fashion and obtain

$$\nabla_{\tilde{\mathbf{x}}_{T,<}} \left[ \log p\left(\left(\mathbf{U}^T\mathbf{z} - \mathbf{\Sigma}\mathbf{V}^T\mathbf{n}\right)_< | \tilde{\mathbf{x}}_T\right) + \log p\left(\left(\mathbf{U}^T\mathbf{z} - \mathbf{\Sigma}\mathbf{V}^T\mathbf{n}\right)_{\not<} | \tilde{\mathbf{x}}_T\right) \right] =$$
$$= \nabla_{\tilde{\mathbf{x}}_{T,<}} \log p\left(\left(\mathbf{U}^T\mathbf{z} - \mathbf{\Sigma}\mathbf{V}^T\mathbf{n}\right)_< | \tilde{\mathbf{x}}_T\right) = \nabla_{\tilde{\mathbf{x}}_{T,<}} \log p\left(\left(\mathbf{y}_T - \mathbf{\Sigma}\tilde{\mathbf{x}}_T\right)_< | \tilde{\mathbf{x}}_T\right).$$

The vector $\left(\mathbf{U}^T\mathbf{z} - \mathbf{\Sigma}\mathbf{V}^T\mathbf{n}\right)_{\not<}$ was built element-wise as gradual noise additions, independent of $\tilde{\mathbf{x}}_{T,<}$, and thus its derivative is zero. We obtain a gradient-log of a Gaussian density function of the vector $(\mathbf{y}_T - \mathbf{\Sigma}\tilde{\mathbf{x}}_T)_<$, having a zero mean and a covariance matrix $\sigma_0^2\mathbf{I} - \sigma_i^2\mathbf{\Sigma}_<\mathbf{\Sigma}_<^T$, according to Equation 2. Thus, when deriving it by $\tilde{\mathbf{x}}_{T,<}$, we obtain the known Gaussian gradient-log, multiplied from the left by $-\mathbf{\Sigma}_<^T$, which is the inner derivative of the Gaussian parameter, implying

$$\nabla_{\tilde{\mathbf{x}}_{T,<}} \log p\left(\tilde{\mathbf{x}}_T | \mathbf{y}_T\right) = -\mathbf{\Sigma}_<^T \left(\sigma_0^2\mathbf{I} - \sigma_i^2\mathbf{\Sigma}_<\mathbf{\Sigma}_<^T\right)^{-1} \left(\mathbf{\Sigma}\tilde{\mathbf{x}}_T - \mathbf{y}_T\right)_< + \left(\nabla_{\tilde{\mathbf{x}}_T} \log p\left(\tilde{\mathbf{x}}_T\right)\right)_<$$
$$= \mathbf{\Sigma}_<^T \left(\sigma_0^2\mathbf{I} - \sigma_i^2\mathbf{\Sigma}_<\mathbf{\Sigma}_<^T\right)^{-1} \left(\mathbf{y}_T - \mathbf{\Sigma}\tilde{\mathbf{x}}_T\right)_< + \left(\nabla_{\tilde{\mathbf{x}}_T} \log p\left(\tilde{\mathbf{x}}_T\right)\right)_<$$
$$= \left(\mathbf{\Sigma}^T \left(\sigma_0^2\mathbf{I} - \sigma_i^2\mathbf{\Sigma}\mathbf{\Sigma}^T\right)^{-1} \left(\mathbf{y}_T - \mathbf{\Sigma}\tilde{\mathbf{x}}_T\right)\right)_< + \left(\mathbf{V}^T\nabla_{\tilde{\mathbf{x}}} \log p\left(\tilde{\mathbf{x}}\right)\right)_< .$$

So, in summary,

$$\nabla_{\tilde{\mathbf{x}}_{T,<}} \log p\left(\tilde{\mathbf{x}}_T | \mathbf{y}_T\right) = \left(\mathbf{\Sigma}^T \left(\sigma_0^2\mathbf{I} - \sigma_i^2\mathbf{\Sigma}\mathbf{\Sigma}^T\right)^{-1} \left(\mathbf{y}_T - \mathbf{\Sigma}\tilde{\mathbf{x}}_T\right)\right)_< + \left(\mathbf{V}^T\nabla_{\tilde{\mathbf{x}}} \log p\left(\tilde{\mathbf{x}}\right)\right)_< . \quad (9)$$

Aggregating all these results together, by combining Equations 4, 8 and 9 into one vector, we obtain the following conditional score function approximation:

$$\nabla_{\tilde{\mathbf{x}}_T} \log p\left(\tilde{\mathbf{x}}_T | \mathbf{y}_T\right) = \mathbf{\Sigma}^T \left|\sigma_0^2\mathbf{I} - \sigma_i^2\mathbf{\Sigma}\mathbf{\Sigma}^T\right|^\dagger \left(\mathbf{y}_T - \mathbf{\Sigma}\tilde{\mathbf{x}}_T\right) + \left.\left(\mathbf{V}^T\nabla_{\tilde{\mathbf{x}}} \log p\left(\tilde{\mathbf{x}}\right)\right)\right|_{\not>}, \quad (10)$$

where $(\mathbf{v})|_{\not>}$ is the vector $\mathbf{v}$, but with zeros in its entries that correspond to $\mathbf{v}_>$. Observe that the first term in Equation 10 contains zeros in the entries corresponding to $\tilde{\mathbf{x}}_{T,0}$, matching the above calculations.

∎

# B   Step Size Derivation

As explained in [1], the following equality holds:

$$\nabla_{\tilde{\mathbf{x}}} \log p\left(\tilde{\mathbf{x}}\right) = \frac{\mathbf{D}\left(\tilde{\mathbf{x}}, \sigma\right) - \tilde{\mathbf{x}}}{\sigma^2},$$

where $\mathbf{D}\left(\tilde{\mathbf{x}}, \sigma\right)$ is the theoretical MSE minimizer, $\mathbb{E}\left[\mathbf{x} | \tilde{\mathbf{x}}\right]$. We introduce an assumption that $\mathbf{D}\left(\tilde{\mathbf{x}}, \sigma\right)$ does not significantly change with small perturbations in $\tilde{\mathbf{x}}$, resulting in:

$$\frac{\partial}{\partial \tilde{\mathbf{x}}} \mathbf{D}\left(\tilde{\mathbf{x}}, \sigma\right) \approx \mathbf{0}.$$

This assumption is justified by the fact that with probability 1, the infinitesimal perturbations are orthogonal to the image manifold around the point $\tilde{\mathbf{x}}$, implying that they can be referred to as an additive white Gaussian noise. Due to the efficiency of the denoiser in wiping such noise, the sensitivity of its output to this extra noise is negligible.

Our goal in this appendix is to evaluate the Hessian of the log posterior in order to be used for better conditioning of the iterative Langevin steps. Thus, we need to differentiate the gradient that was derived above,

$$\nabla_{\tilde{\mathbf{x}}_T} \log p\left(\tilde{\mathbf{x}}_T | \mathbf{y}_T\right) = \mathbf{\Sigma}^T \left|\sigma_0^2\mathbf{I} - \sigma_i^2\mathbf{\Sigma}\mathbf{\Sigma}^T\right|^\dagger \left(\mathbf{y}_T - \mathbf{\Sigma}\tilde{\mathbf{x}}_T\right) + \left.\left(\mathbf{V}^T\nabla_{\tilde{\mathbf{x}}} \log p\left(\tilde{\mathbf{x}}\right)\right)\right|_{\not>}.$$

**Theorem 2.** *Given* $\mathbf{y} = \mathbf{Hx} + \mathbf{z}$, $\mathbf{z} \sim \mathcal{N}\left(0, \sigma_0^2 \mathbf{I}\right)$, $\mathbf{H} = \mathbf{U}\mathbf{\Sigma}\mathbf{V}^T$ *is the SVD decomposition of* $\mathbf{H}$, $\mathbf{y}_T = \mathbf{U}^T\mathbf{y}$, $\mathbf{n} = \mathbf{n}_i$ *as constructed in subsection 3.1 in the main paper,* $\tilde{\mathbf{x}} = \tilde{\mathbf{x}}_i = \mathbf{x} + \mathbf{n}$, $\tilde{\mathbf{x}}_T = \mathbf{V}^T\tilde{\mathbf{x}}$, $\mathbf{x}_T = \mathbf{V}^T\mathbf{x}$, *the Hessian of the log posterior can be approximated by a diagonal matrix whose entries are:*

$$\left[\nabla^2_{\tilde{\mathbf{x}}_T} \log p\left(\tilde{\mathbf{x}}_T | \mathbf{y}_T\right)\right]_{j,j} = \begin{cases} \frac{-1}{\sigma_i^2} & s_j = 0 \\ \frac{-s_j^2}{s_j^2 \sigma_i^2 - \sigma_0^2} & \sigma_i s_j > \sigma_0 \\ \frac{-s_j^2}{\sigma_0^2 - s_j^2 \sigma_i^2} - \frac{1}{\sigma_i^2} & 0 < \sigma_i s_j < \sigma_0. \end{cases}$$

*Proof.* Again, we split our calculation into 3 cases:

**For the case of** $\tilde{\mathbf{x}}_{T,0}$, we notice that the first term in the gradient is zero due to the multiplication by $\mathbf{\Sigma}^T$, and thus we calculate:

$$\nabla^2_{\tilde{\mathbf{x}}_{T,0}} \log p\left(\tilde{\mathbf{x}}_T | \mathbf{y}_T\right) = \frac{\partial}{\partial \tilde{\mathbf{x}}_{T,0}} \left(\mathbf{V}^T \nabla_{\tilde{\mathbf{x}}} \log p\left(\tilde{\mathbf{x}}\right)\right)\big|_{\not\tilde{\mathbf{x}}}.$$

We use the chain rule and obtain

$$\begin{aligned} \nabla^2_{\tilde{\mathbf{x}}_{T,0}} \log p\left(\tilde{\mathbf{x}}_T | \mathbf{y}_T\right) &= \mathbf{V}_0 \frac{\partial}{\partial \tilde{\mathbf{x}}_0} \left(\mathbf{V}^T \nabla_{\tilde{\mathbf{x}}} \log p\left(\tilde{\mathbf{x}}\right)\right)\big|_{\not\tilde{\mathbf{x}}} \\ &= \mathbf{V}_0 \mathbf{V}_0^T \frac{\partial}{\partial \tilde{\mathbf{x}}_0} \left(\nabla_{\tilde{\mathbf{x}}} \log p\left(\tilde{\mathbf{x}}\right)\right)\big|_{\not\tilde{\mathbf{x}}} \\ &= \frac{\partial}{\partial \tilde{\mathbf{x}}_0} \left(\nabla_{\tilde{\mathbf{x}}} \log p\left(\tilde{\mathbf{x}}\right)\right)\big|_{\not\tilde{\mathbf{x}}} \\ &= \frac{\partial}{\partial \tilde{\mathbf{x}}_0} \left(\frac{\mathbf{D}\left(\tilde{\mathbf{x}}, \sigma\right) - \tilde{\mathbf{x}}}{\sigma_i^2}\right)\Big|_{\not\tilde{\mathbf{x}}} \\ &= \frac{\partial}{\partial \tilde{\mathbf{x}}_0} \left(\frac{-\tilde{\mathbf{x}}}{\sigma_i^2}\right)\Big|_{\not\tilde{\mathbf{x}}} \\ &= \frac{-1}{\sigma_i^2} \mathbf{I}, \end{aligned}$$

where we have invoked our earlier assumption on the denoiser's sensitivity to perturbations. This leads to the conclusion

$$\nabla^2_{\tilde{\mathbf{x}}_{T,0}} \log p\left(\tilde{\mathbf{x}}_T | \mathbf{y}_T\right) = \frac{-1}{\sigma_i^2} \mathbf{I}. \tag{11}$$

**For the case of** $\tilde{\mathbf{x}}_{T,>}$, we calculate:

$$\nabla^2_{\tilde{\mathbf{x}}_{T,>}} \log p\left(\tilde{\mathbf{x}}_T | \mathbf{y}_T\right) = \frac{\partial}{\partial \tilde{\mathbf{x}}_{T,>}} \left[\mathbf{\Sigma}^T \left|\sigma_0^2 \mathbf{I} - \sigma_i^2 \mathbf{\Sigma}\mathbf{\Sigma}^T\right|^\dagger \left(\mathbf{y}_T - \mathbf{\Sigma}\tilde{\mathbf{x}}_T\right) + \left(\mathbf{V}^T \nabla_{\tilde{\mathbf{x}}} \log p\left(\tilde{\mathbf{x}}\right)\right)\big|_{\not\tilde{\mathbf{x}}}\right].$$

The first term's derivative is simply the matrix that multiplies the vector $\tilde{\mathbf{x}}_{T,>}$, while the second term can be approximated, with the use of the chain rule, as follows:

$$\begin{aligned} \frac{\partial}{\partial \tilde{\mathbf{x}}_{T,>}} \left(\mathbf{V}^T \nabla_{\tilde{\mathbf{x}}} \log p\left(\tilde{\mathbf{x}}\right)\right)\big|_{\not\tilde{\mathbf{x}}} &= \mathbf{V}_> \frac{\partial}{\partial \tilde{\mathbf{x}}_>} \left(\mathbf{V}^T \nabla_{\tilde{\mathbf{x}}} \log p\left(\tilde{\mathbf{x}}\right)\right)\big|_{\not\tilde{\mathbf{x}}} \\ &= \mathbf{V}_> \mathbf{V}_>^T \frac{\partial}{\partial \tilde{\mathbf{x}}_>} \left(\nabla_{\tilde{\mathbf{x}}} \log p\left(\tilde{\mathbf{x}}\right)\right)\big|_{\not\tilde{\mathbf{x}}} \\ &= \frac{\partial}{\partial \tilde{\mathbf{x}}_>} \left(\nabla_{\tilde{\mathbf{x}}} \log p\left(\tilde{\mathbf{x}}\right)\right)\big|_{\not\tilde{\mathbf{x}}} \\ &= \frac{\partial}{\partial \tilde{\mathbf{x}}_>} \left(\frac{\mathbf{D}\left(\tilde{\mathbf{x}}, \sigma\right) - \tilde{\mathbf{x}}}{\sigma_i^2}\right)\Big|_{\not\tilde{\mathbf{x}}} \\ &= \frac{\partial}{\partial \tilde{\mathbf{x}}_>} \left(\frac{-\tilde{\mathbf{x}}}{\sigma_i^2}\right)\Big|_{\not\tilde{\mathbf{x}}} \\ &= \mathbf{0}, \end{aligned}$$

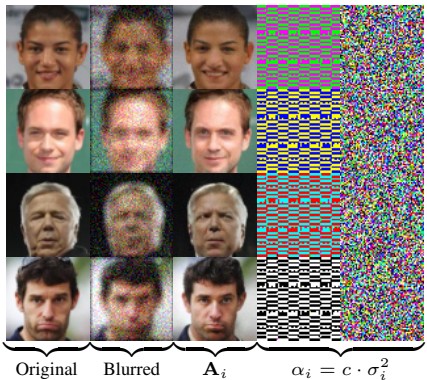

Original   Blurred   $\mathbf{A}_i$   $\alpha_i = c \cdot \sigma_i^2$

Figure 1: Comparison of different step sizes, while the rest of the hyperparameters are fixed (uniform $5 \times 5$ blur and an additive noise with $\sigma_0 = 0.1$). The third column refers to the diagonal step size matrix $\mathbf{A}_i$, as used in SNIPS. The last two columns refer to a uniform time-dependent step size $\alpha_i = c \cdot \sigma_i^2$, with $c = 1e-3, 1e-5$, respectively. Different choices of $c$ yielded similar results.

where we have invoked our earlier assumption on the denoiser's sensitivity to perturbations, resulting in

$$\nabla^2_{\tilde{\mathbf{x}}_{T,>}} \log p\left(\tilde{\mathbf{x}}_T | \mathbf{y}_T\right) = \left(-\boldsymbol{\Sigma}^T \left|\sigma_0^2 \mathbf{I} - \sigma_i^2 \boldsymbol{\Sigma} \boldsymbol{\Sigma}^T\right|^\dagger \boldsymbol{\Sigma}\right)_{>}. \tag{12}$$

**For the case of $\tilde{\mathbf{x}}_{T,<}$**, we calculate:

$$\nabla^2_{\tilde{\mathbf{x}}_{T,<}} \log p\left(\tilde{\mathbf{x}}_T | \mathbf{y}_T\right) = \frac{\partial}{\partial \tilde{\mathbf{x}}_{T,<}} \left[\boldsymbol{\Sigma}^T \left|\sigma_0^2 \mathbf{I} - \sigma_i^2 \boldsymbol{\Sigma} \boldsymbol{\Sigma}^T\right|^\dagger (\mathbf{y}_T - \boldsymbol{\Sigma} \tilde{\mathbf{x}}_T) + \left(\mathbf{V}^T \nabla_{\tilde{\mathbf{x}}} \log p\left(\tilde{\mathbf{x}}\right)\right)\big|_{\not>}\right].$$

The first term's derivative can be calculated similarly to the previous case, and the second term can be approximately derived as in the first case, resulting in

$$\nabla^2_{\tilde{\mathbf{x}}_{T,<}} \log p\left(\tilde{\mathbf{x}}_T | \mathbf{y}_T\right) = \left(-\boldsymbol{\Sigma}^T \left|\sigma_0^2 \mathbf{I} - \sigma_i^2 \boldsymbol{\Sigma} \boldsymbol{\Sigma}^T\right|^\dagger \boldsymbol{\Sigma}\right)_{<} + \frac{-1}{\sigma_i^2}\mathbf{I}. \tag{13}$$

Aggregating all these results together, by combining Equations 11, 12 and 13 into one diagonal matrix, we obtain the following diagonal entries of the Hessian:

$$\left[\nabla^2_{\tilde{\mathbf{x}}_T} \log p\left(\tilde{\mathbf{x}}_T | \mathbf{y}_T\right)\right]_{j,j} = \begin{cases} \frac{-1}{\sigma_i^2} & s_j = 0 \\ \frac{-s_j^2}{s_j^2 \sigma_i^2 - \sigma_0^2} & \sigma_i s_j > \sigma_0 \\ \frac{-s_j^2}{\sigma_0^2 - s_j^2 \sigma_i^2} - \frac{1}{\sigma_i^2} & 0 < \sigma_i s_j < \sigma_0. \end{cases} \tag{14}$$

∎

Finally, since the approximation of the Hessian is a diagonal matrix and its diagonal entries are non-zeros, we can easily invert it. This results in the following term for each of the diagonal entries of the negative inverse Hessian, which we denote $\mathbf{A}_i$:

$$\left(\mathbf{A}_i\right)_{j,j} = \begin{cases} \sigma_i^2 & s_j = 0 \\ \sigma_i^2 - \frac{\sigma_0^2}{s_j^2} & \sigma_i s_j > \sigma_0 \\ \sigma_i^2 \cdot \left(1 - s_j^2 \frac{\sigma_i^2}{\sigma_0^2}\right) & 0 < \sigma_i s_j < \sigma_0. \end{cases}$$

In order to demonstrate the effectiveness of this position-dependent step size vector, we compare it to a uniform step size $\alpha_i \propto \sigma_i^2$ for image deblurring. As can be seen in Figure 1, the latter diverges under the same hyperparameters. It is possible that for a large enough number of iterations, a uniform step size might converge and produce viable results. However, we find little value in demonstrating this, as it requires retraining the NCSNv2 model for more noise levels, and it slows down the algorithm.

Table 1: Hyperparameters for our experiments. $\frac{\sigma_{i+1}}{\sigma_i}$ is the geometric common ratio for $\{\sigma_i\}_{i=1}^L$.

| Dataset | $c$ | $\tau$ | $L$ | $\sigma_1$ | $\sigma_L$ | $\frac{\sigma_{i+1}}{\sigma_i}$ |
|---|---|---|---|---|---|---|
| **CelebA** | $3.3e-2$ | 5 | 500 | 90 | 0.01 | 0.982 |
| **LSUN** | $1.8e-2$ | 3 | 1086 | 190 | 0.01 | 0.991 |

## C  Alternative Definition of the Noise

In our derivations in subsection 3.1 in the main paper we argued that for the analysis to go through, we should tie the synthetic annealed Langevin noise to the measurements one. As can be seen in Appendix A, this choice clearly complicates the derivation of the conditional score, raising the question whether a simple independence between these two random vectors could have been used instead. In this appendix we explore this option and expose its limitation.

We start by defining $\tilde{\mathbf{x}}_{L+1} = \mathbf{x}$, and for every $i = L, L-1, \ldots, 1$: $\tilde{\mathbf{x}}_i = \tilde{\mathbf{x}}_{i+1} + \boldsymbol{\eta}_i$, where $\boldsymbol{\eta}_i \sim \mathcal{N}\left(0, \left(\sigma_i^2 - \sigma_{i+1}^2\right)\mathbf{I}\right)$ is independent of $\mathbf{z}$. This results in $\tilde{\mathbf{x}}_i = \mathbf{x} + \mathbf{n}_i$, where $\mathbf{n}_i = \sum_{k=i}^L \boldsymbol{\eta}_k \sim \mathcal{N}\left(0, \sigma_i^2\mathbf{I}\right)$. As before, we aim to derive the conditional score function $p\left(\tilde{\mathbf{x}}_i|\mathbf{y}\right)$ and thus we look at the vector

$$\mathbf{y} - \mathbf{H}\tilde{\mathbf{x}}_i = \mathbf{H}\mathbf{x} + \mathbf{z} - \mathbf{H}\mathbf{x} - \mathbf{H}\mathbf{n}_i = \mathbf{z} - \mathbf{H}\mathbf{n}_i. \tag{15}$$

This is a Gaussian vector with zero mean and a covariance matrix $\sigma_0^2\mathbf{I} + \sigma_i^2\mathbf{H}\mathbf{H}^T$, due to the independence between $\mathbf{z}$ and $\mathbf{n}$. In order to make use of Equation 15, we would like to express $p\left(\tilde{\mathbf{x}}_i|\mathbf{y}\right)$ as $p\left(\mathbf{H}\tilde{\mathbf{x}}_i - \mathbf{y}|\mathbf{y}\right)$. However, this transition is not possible because the multiplication by $\mathbf{H}$ is not an invertible operation, which means that it changes the statistics of the tested vector. Instead, $p\left(\tilde{\mathbf{x}}_i|\mathbf{y}\right)$ may be expressed using the Bayes rule as

$$p\left(\tilde{\mathbf{x}}_i|\mathbf{y}\right) = \frac{1}{p\left(\mathbf{y}\right)}p\left(\tilde{\mathbf{x}}_i\right)p\left(\mathbf{y}|\tilde{\mathbf{x}}_i\right) = \frac{1}{p\left(\mathbf{y}\right)}p\left(\tilde{\mathbf{x}}_i\right)p\left(\mathbf{y} - \mathbf{H}\tilde{\mathbf{x}}_i|\tilde{\mathbf{x}}_i\right).$$

The first term $1/p\left(\mathbf{y}\right)$ becomes zero after differentiating by $\tilde{\mathbf{x}}_i$, and the second term's gradient log can be approximated using a neural network, as done before. The third term describes a Gaussian vector, and can be written as $p\left(\mathbf{z} - \mathbf{H}\mathbf{n}_i|\tilde{\mathbf{x}}_i\right)$ due to Equation 15. The Gaussian vector $\mathbf{z} - \mathbf{H}\mathbf{n}_i$ is conditioned on $\tilde{\mathbf{x}}_i = \mathbf{x}_i + \mathbf{n}_i$, which encapsulates information about $\mathbf{n}_i$, without a clear way of knowing $\mathbf{n}_i$ itself. Thus, without an explicit term for $p\left(\mathbf{n}_i|\tilde{\mathbf{x}}_i\right)$, we are unable to derive an analytical term for the gradient log of the likelihood.

Therefore, the path we took to define the noise additions aims for the difference $\mathbf{y} - \mathbf{H}\tilde{\mathbf{x}}_i$ to be independent of $\tilde{\mathbf{x}}_i$. In order to achieve that, we use the SVD decomposition of $\mathbf{H}$ and define the noise addition sequence as in subsection 3.1 in the main paper, both steps seem unavoidable.

## D  Implementation Details

We run SNIPS with the hyperparameters detailed in Table 1, where $\{\sigma_i\}_{i=1}^L$ is a decreasing geometric sequence. These hyperparameters conform to those used in NCSNv2 [3], the neural network model that we used. The parameters $\mathbf{H}$, $\sigma_0$ and $\mathbf{y}$ are defined by the inverse problem at hand. Recall that this algorithm applies $\tau L$ overall iterations to complete, in each a denoiser is being activated. The sampling algorithm was run on a single Nvidia RTX3080 GPU with 10GB memory, and took around 2 minutes for producing 8 samples from the $64 \times 64$ CelebA dataset, and around 6 minutes for producing 6 samples from the $128 \times 128$ LSUN dataset. The exact times vary slightly for the various inverse problems.

The code used in this paper is available at `https://github.com/bahjat-kawar/snips_torch`.

## E  Comparison to RED

RED [2] is a well-known method that leverages a denoiser for the MAP solution of inverse problems, and as such it is a relevant method to compare with. We compare RED to SNIPS on the image

Table 2: Comparison between SNIPS and RED on 8 CelebA images. SNIPS Mean is the average of 8 SNIPS outputs per image. The best number in each row is in **bold**.

| Problem | Metric | SNIPS | SNIPS Mean | RED |
|---|---|---|---|---|
| Uniform deblurring with $\sigma_0 = 0.006$ | PSNR $\uparrow$ | 32.41 | **35.42** | 29.03 |
| | LPIPS $\downarrow$ | **0.005** | **0.005** | 0.043 |
| Uniform deblurring with $\sigma_0 = 0.1$ | PSNR $\uparrow$ | 25.03 | **27.28** | 20.10 |
| | LPIPS $\downarrow$ | **0.032** | 0.045 | 0.077 |

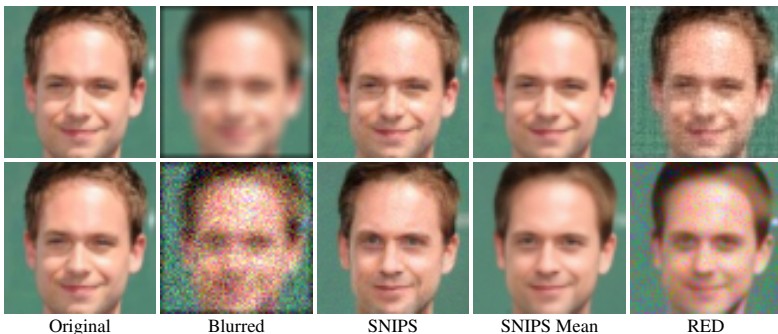

| Original | Blurred | SNIPS | SNIPS Mean | RED |

Figure 2: Deblurring results on a CelebA image (uniform $5 \times 5$ blur). Top: additive noise with $\sigma_0 = 0.006$, bottom: additive noise with $\sigma_0 = 0.1$.

deblurring problem (with a uniform $5 \times 5$ kernel and additive noise with $\sigma_0$), while using the same denoiser model (NCSNv2) for both. We run the SD (Steepest Descent) version of RED on the luminance channel of the image in the YCbCr color space, as in the original paper, with its hyperparameters chosen for best PSNR performance. Namely, $\lambda = 0.12$, $N = 100$ for $\sigma_0 = 0.006$, and $\lambda = 1000$, $N = 100$ for $\sigma_0 = 0.1$. In addition to PSNR, we also calculate LPIPS [4], a perceptual quality metric, in order to verify the claim that SNIPS has superior visual quality.

As can be seen in Table 2, both SNIPS and its mean outperform RED in PSNR as well as LPIPS. When the noise is significant ($\sigma_0 = 0.1$), it becomes clear that SNIPS has superior visual quality at the expense of PSNR performance, in comparison to the average of samples. A visual comparison is shown in Figure 2.

## F   Additional Results

We provide below more results of SNIPS for image deblurring, super-resolution and compressive sampling. We recommend to view these figures zoomed-in in order to see the details in the produced samples (or lack thereof in their average).

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

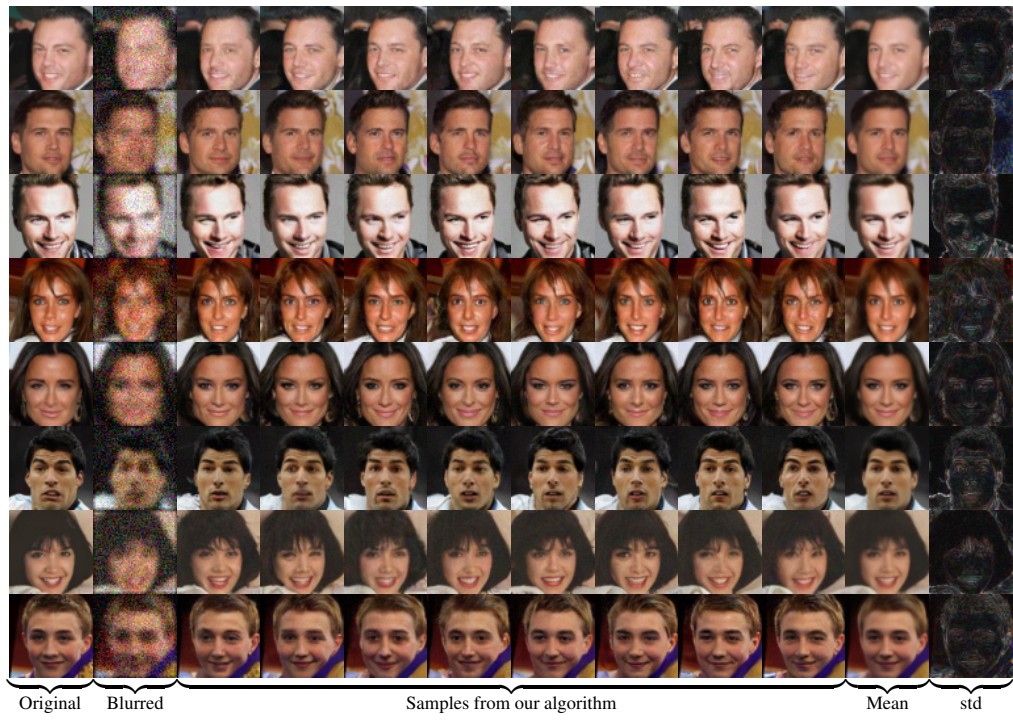

Original  Blurred                Samples from our algorithm                    Mean   std

Figure 3: Deblurring results on CelebA images (uniform $5 \times 5$ blur and an additive noise with $\sigma_0 = 0.1$).

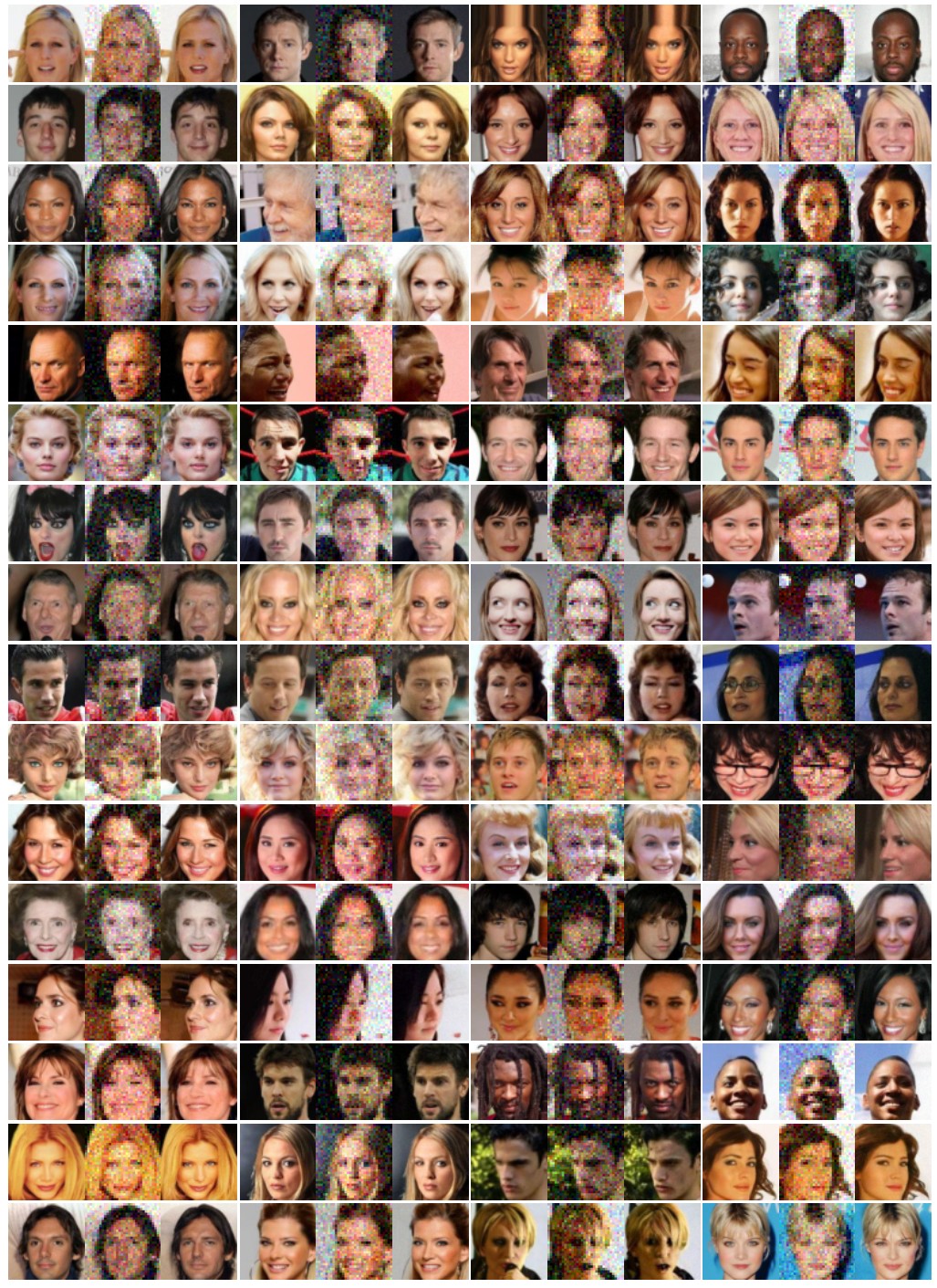

Figure 4: Extended uncurated super resolution results on CelebA images (downscaling $2:1$ by plain averaging and adding noise with $\sigma_0 = 0.1$). Every image set contains: original, low-res, SNIPS restoration, in that order.

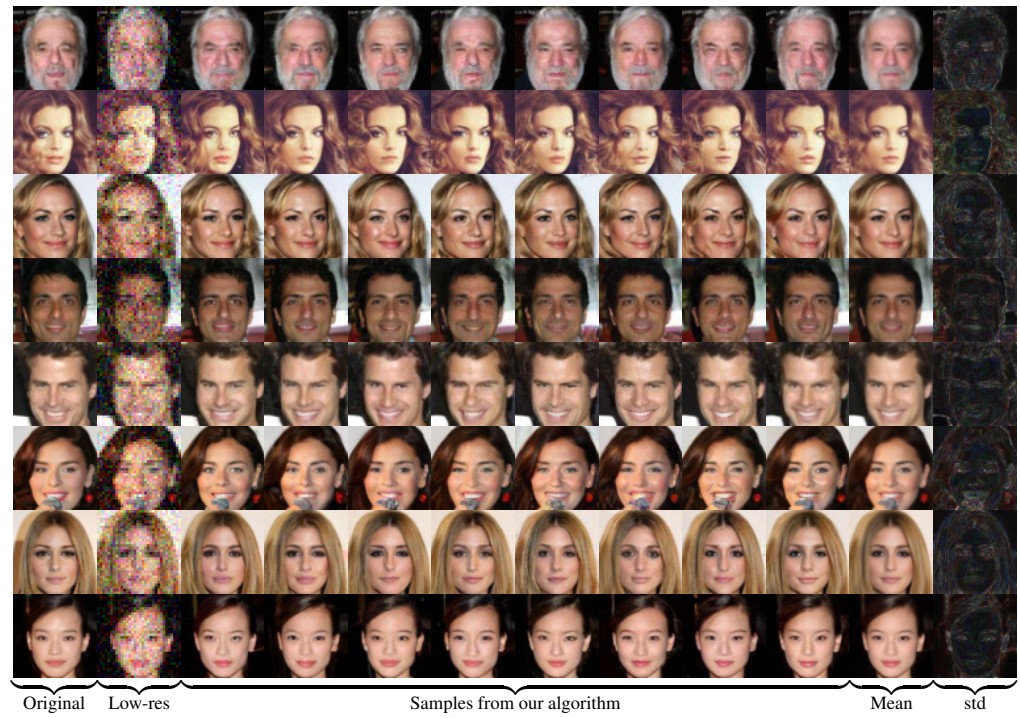

Original    Low-res                Samples from our algorithm                Mean    std

Figure 5: Super resolution results on CelebA images (downscaling $2:1$ by plain averaging and adding noise with $\sigma_0 = 0.1$).

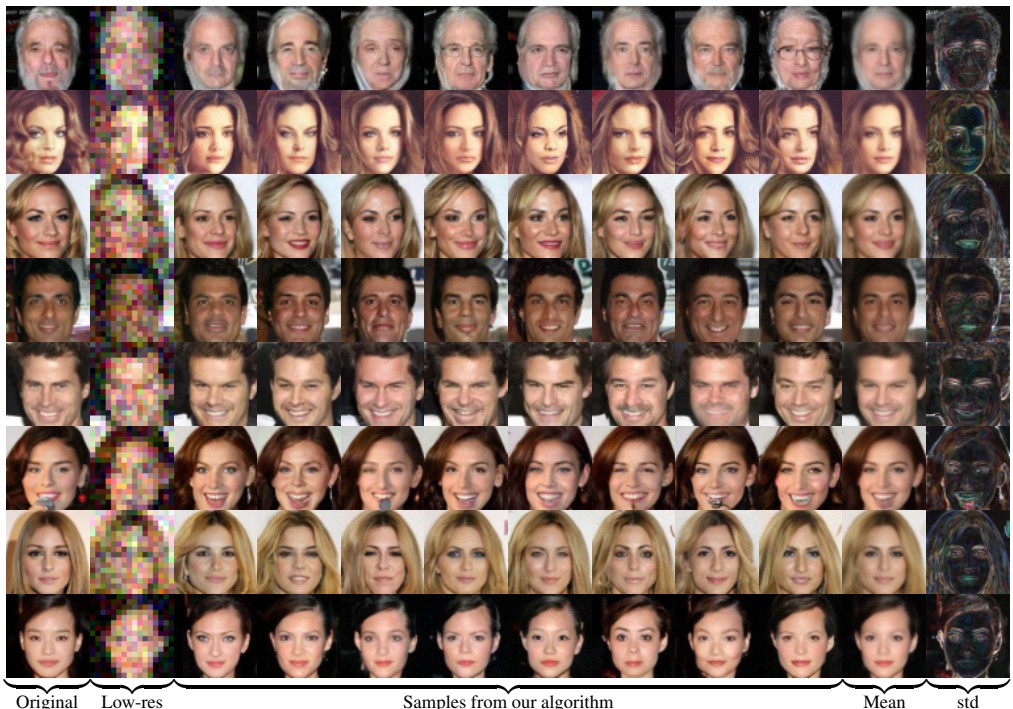

Original    Low-res                Samples from our algorithm                Mean    std

Figure 6: Super resolution results on CelebA images (downscaling $4:1$ by plain averaging and adding noise with $\sigma_0 = 0.1$).

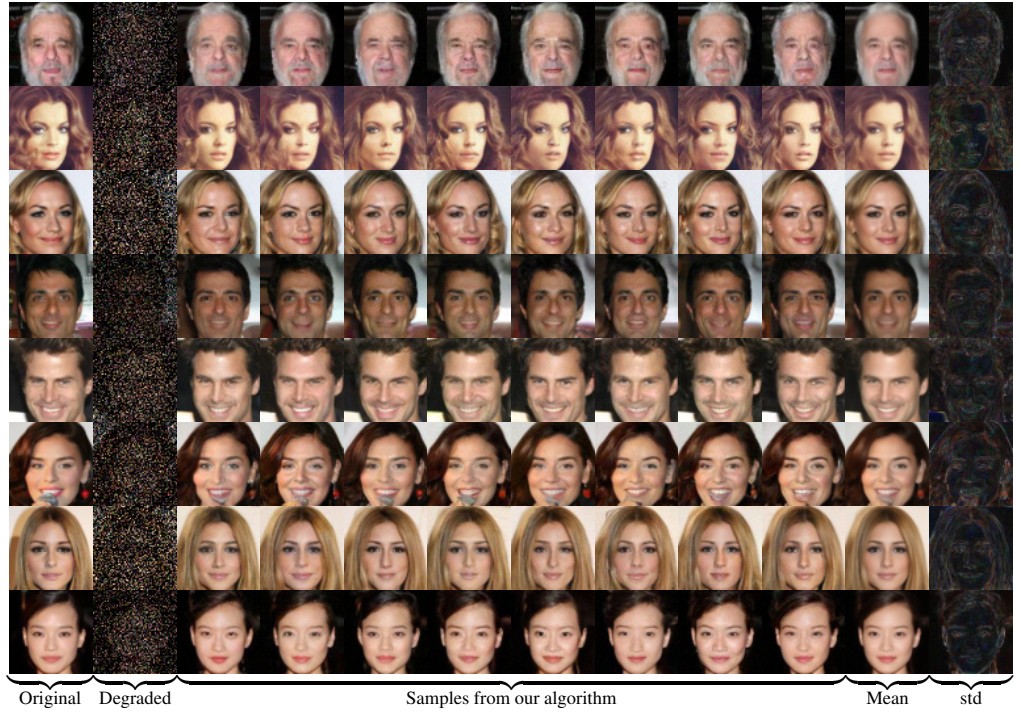

Original  Degraded          Samples from our algorithm          Mean      std

Figure 7: Compressive sensing results on CelebA images (compression by $25\%$ and adding noise with $\sigma_0 = 0.1$).

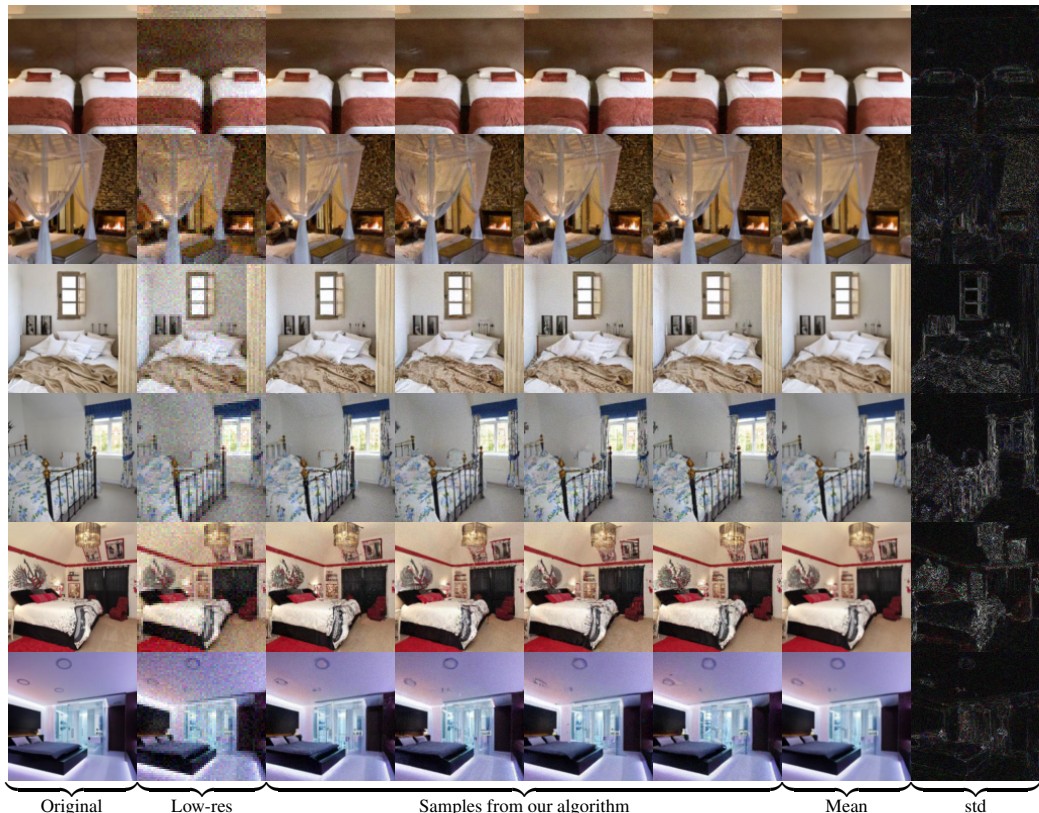

Original    Low-res          Samples from our algorithm          Mean      std

Figure 8: Super resolution results on LSUN bedroom images (downscaling $2 : 1$ by plain averaging and adding noise with $\sigma_0 = 0.04$).

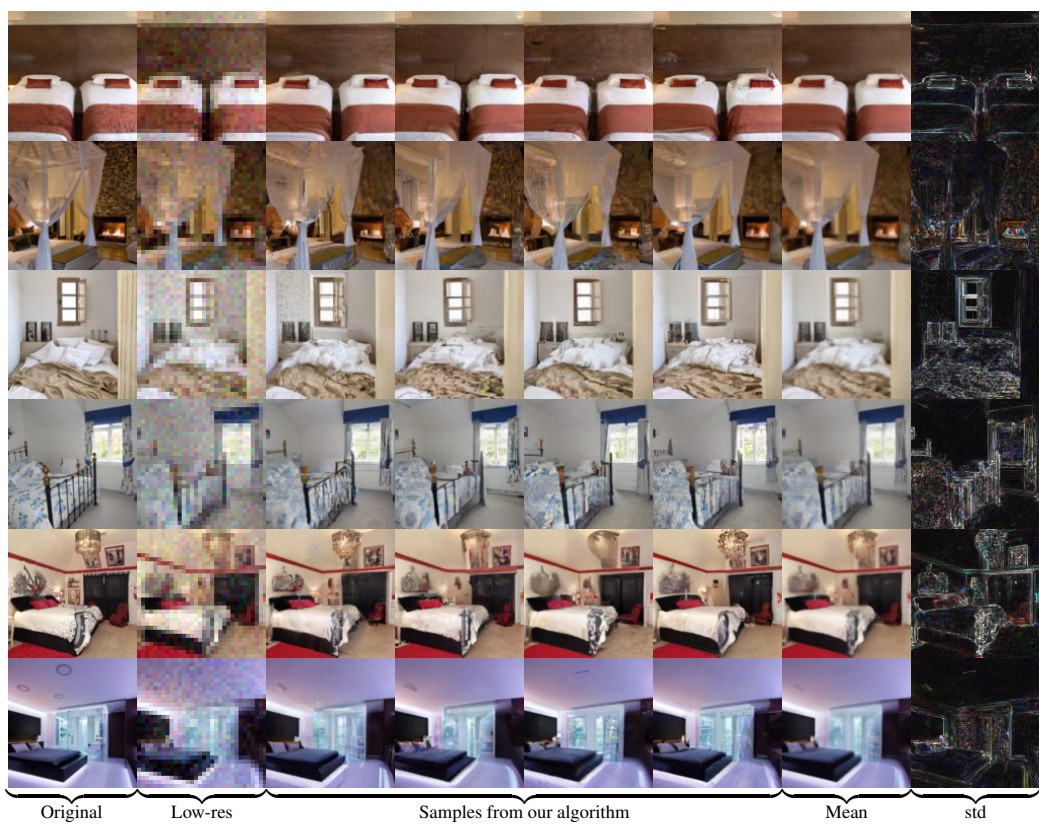

Original    Low-res    Samples from our algorithm    Mean    std

Figure 9: Super resolution results on LSUN bedroom images (downscaling $4:1$ by plain averaging and adding noise with $\sigma_0 = 0.04$).

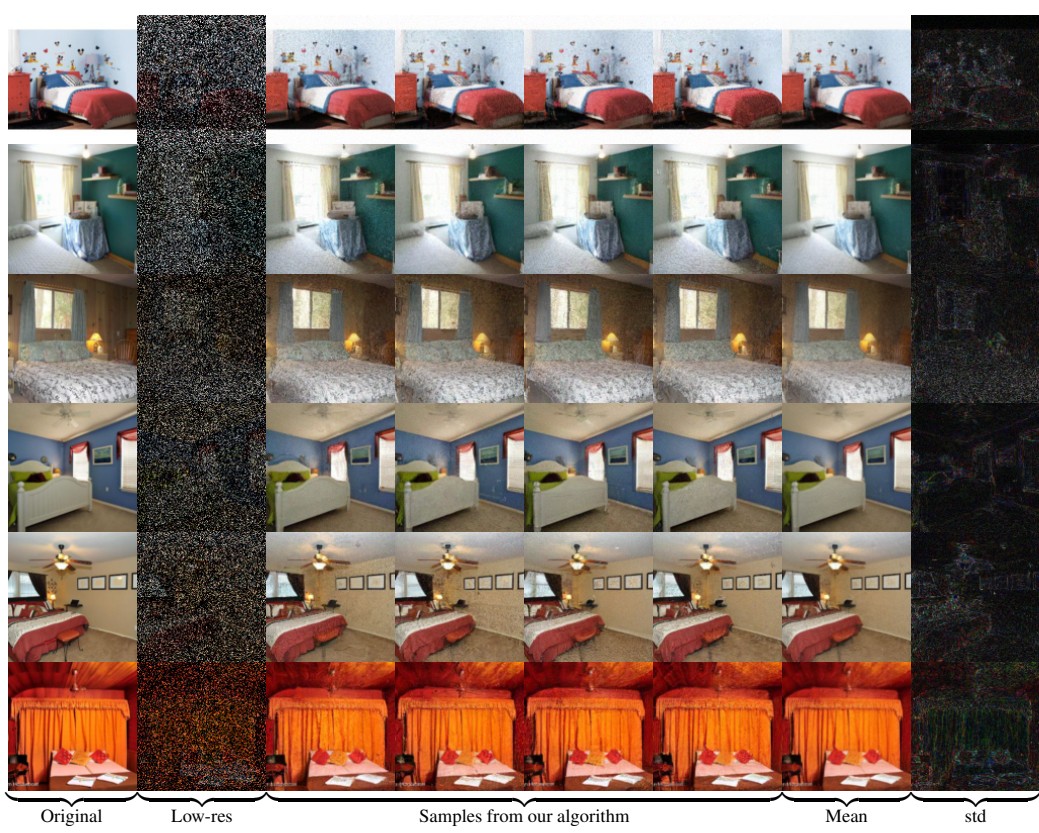

Original    Low-res    Samples from our algorithm    Mean    std

Figure 10: Compressive sensing results on LSUN bedroom images (compression by $25\%$ and adding noise with $\sigma_0 = 0.04$).

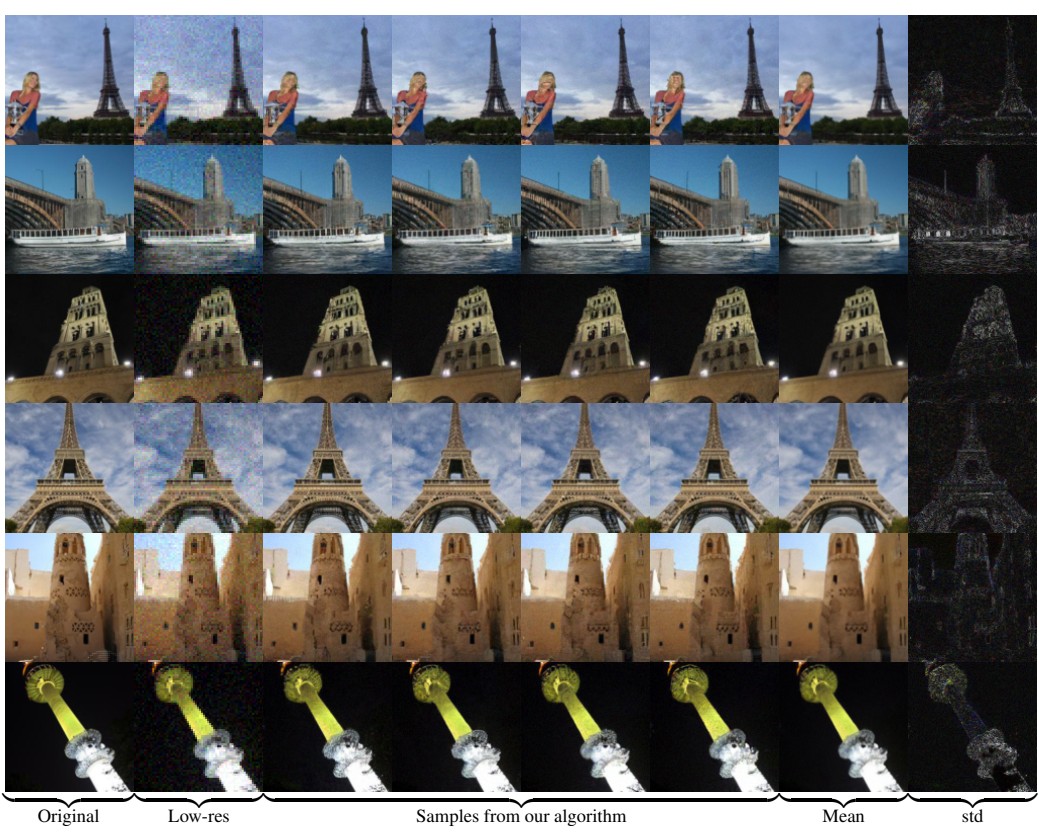

Figure 11: Super resolution results on LSUN tower images (downscaling $2 : 1$ by plain averaging and adding noise with $\sigma_0 = 0.04$).

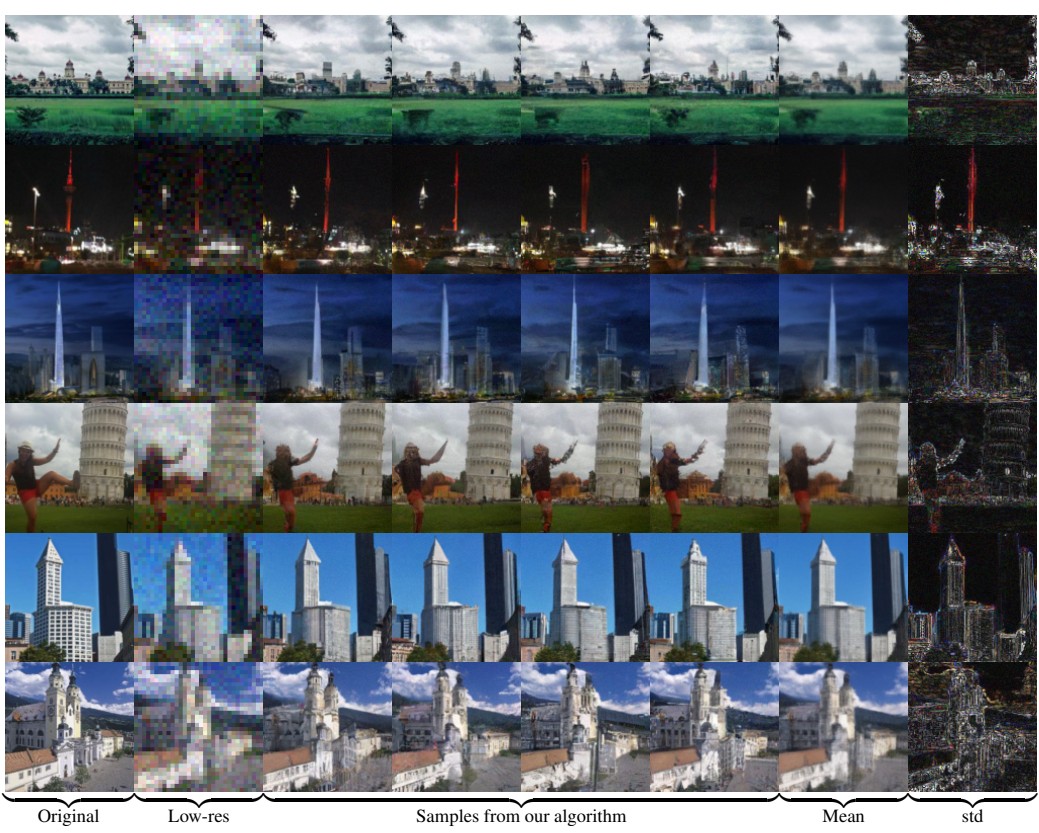

Figure 12: Super resolution results on LSUN tower images (downscaling $4 : 1$ by plain averaging and adding noise with $\sigma_0 = 0.04$).

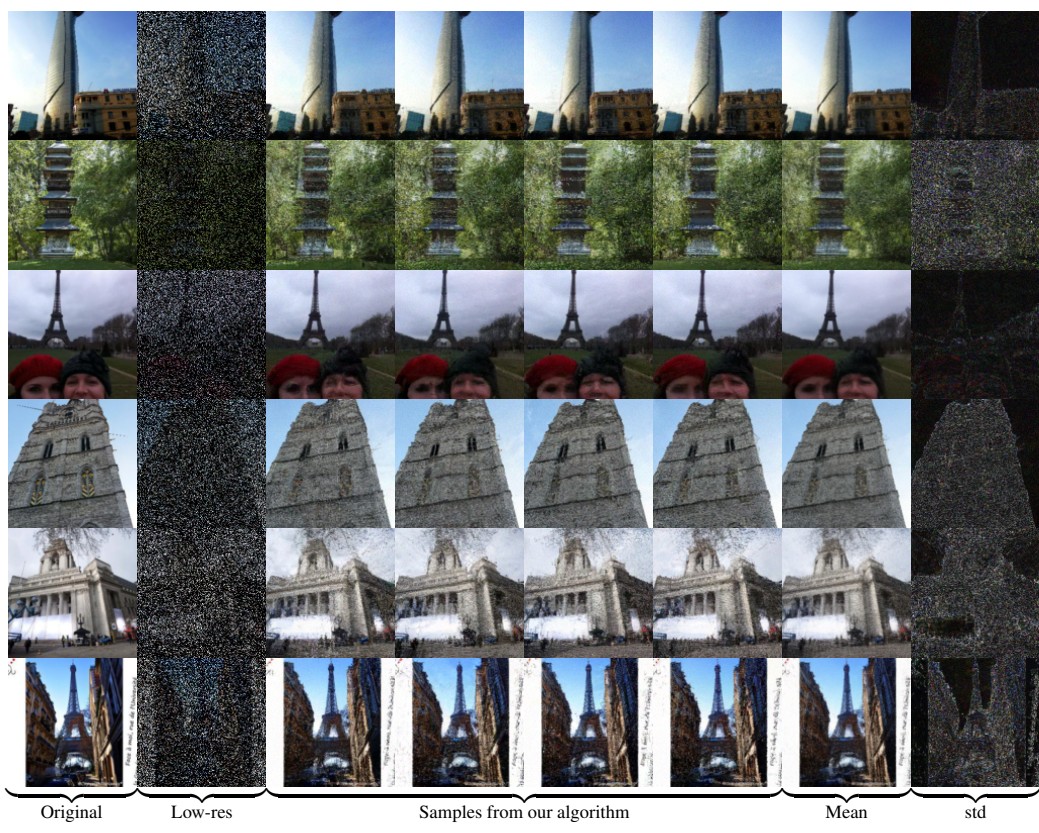

Figure 13: Compressive sensing results on LSUN tower images (compression by $25\%$ and adding noise with $\sigma_0 = 0.04$).