# OpenReview forum: "SNIPS: Solving Noisy Inverse Problems Stochastically"
_NeurIPS.cc/2021/Conference — NeurIPS 2021 Poster_

### Official Review · Reviewer_d1T5 · 2021-07-15

**Rating:** 5
**Confidence:** 3

**Summary:**

This manuscript describes a novel method for sampling from the posterior in a linear inverse problem subjected to noise for an arbitrary measurement matrix. This is achieved by combining a black-box prior (such as that derived from a minimum MSE Gaussian denoiser) whose score function is easily computable with an annealed Langevin dynamics framework. The novelty is the construction of the Langevin noise term, which guarantees that the conditional score function is tractable. In addition, the authors propose a heuristic for step size calculation inspired by Newton's method in optimization. The method is evaluated on a variety of linear inverse problems: image deblurring, super resolution, and compressive sensing. For each task, realistic results are achieved and the residual is shown to conform to the imposed statistical model.


**Ethical Concerns:**

No ethical issues are raised by the manuscript.


**Limitations And Societal Impact:**

The authors discuss the limitations and necessary extensions in Section 6 of their work. These are reasonable and highlight areas of improvement that should be pursued. As this is a theoretical work, there is no expectation of negative societal impact and the authors have accordingly not discussed it.


**Main Review:**

This work proposes a novel and interesting method for the chosen task: sampling from the posterior distribution in a linear inverse problem affected by noise. The motivation of the method and the results are quite compelling. There are, however, some issues with the description of the algorithm that should be addressed. If these changes are made, I would recommend the publication of this manuscript as part of the proceedings. More detailed comments follow below.

First, the convention used for the SVD is a bit non-standard in that the singular value matrix Σ is rectangular (which results in odd expressions such as Σ^T and (Σ^†)^T). I would suggest using the convention that Σ is M×M and V is N×M. Some explanation for the rewrite in eq. (5), where y is written in terms of x-tilde instead of x would also be helpful. Similarly, it would be good to explain why it is desirable that z_T – Σn_T be white and independent of x_T-tilde. Throughout, explanations for various derivations would help the reader to follow the choices made and why. This is especially true for derivations and calculations that are relegated to the supplementary material. Here, it would be good to at least give an intuition for why the result should be true.

The most problematic part is the second half of Section 3.1, the derivation of the annealing noise term. First, the requirement that σ_i s_j ≠ σ_0 for all i, j is a bit worrisome. While it can be easily circumvented by an appropriate choice of variances σ_i, such an inequality constraint suggests an instability in the underlying relationships. The question is what happens when σ_i s_j is close to, but not equal to σ_0. While exact equality is very unlikely (especially given the floating-point representations used in practice), there is nothing preventing these values from being rather close to one another. What would happen if this were the case? Would convergence suffer? Do the calculations come out the same, with factors cancelling out in the limit?

Continuing, the notation i_1 is a bit odd. Where does the subscript one come from? Also, the dependence on j should be made clear, so that i_1 would be come i_1(j) (or simply i_j?). The definition is also only given for s_j ≠ 0. What happens for s_j = 0? This becomes relevant later (line 185) when the special case s_j = 0 is singled out, but this is in fact undefined since i_1 is not given in this case.

The derivation of the noise term itself is quite confusing. First (on line 176), it is stated that η_i is defined as a normally distributed random vector with covariance matrix equal to a multiple of the identity matrix. Then, (on line 182), it is stated that η_{T,i} (the jth component of Σ V^T η_i) is *constructed* to satisfy certain moment conditions. As far as I can tell, what the authors are trying to say here is that η_i has a certain *marginal* distribution, which is a normal distribution with the specified covariance matrix, but that it is also jointly normally distributed with z_T and that the cross-covariance matrix (between η_{T,i} and z_T) is being specified. Nonetheless, this is quite an indirect way of stating the relationship between these random variables.

This becomes more obvious after (on line 183), where the authors state that “this implies that the layers of noise … are all portions of z_T itself”. Again, what I can tease out from this is that the η_{T,i} variables are simply scaled versions of z_T in this case. I think this section would be much better served by explicitly constructing η_i (or η_{T,i}) from z_T (plus an additional independently distributed component). Not only would this be easier to follow, but it would more clearly illustrate the purpose of the construction. The same comment applies to eq. (6). Simply stating that “it can be shown that” (and leaving the calculation to the reader) reduces the clarity of the argument. If the calculations have to be relegated to the supplementary material, at least a brief motivation would be useful here.

Some minor comments follow:

– On line 80, what does it mean for “the entries in the derived score switch mode”? There has been no discussion of modes prior to this.

– It would be helpful for readers less familiar with Langevin dynamics to explain on line 130 why the conditional score function would not be tractable in the general case, since this is a problem that is solved by the proposed algorithm.

– In the problem formulation at the beginning of Section 3.1, it would be good to mention the use of a prior to complete the description of the statistical model (since this is needed to define the posterior).

– The superscript 2 on line 144 is oddly placed, coming right in the middle of some mathematical expressions. It would be better to place it at the end of the sentence.

– In many places (such as lines 179 and 194), the word “white” is used when something more like “heteroscedastic” is meant. A white noise model implies that the components are uncorrelated and that variances of all the components are equal. While this may be the case for η_i, it is not true for Σ V^T η_i. In this case, the covariances are zero, but the variances are different.

– In Sections 3.1 and 3.2, the authors switch between i ≥ i_1, σ_i s_j < σ_0 and s_j < σ_0 / σ_i, all of which are equivalent conditions. I suggest the authors pick one of these characterizations and use it throughout to reduce confusion.

– When giving the step size heuristic in eq. (12), it would be good to provide a brief motivation for why the data term does not enter into the Hessian approximation, since the expression of the score function in eq. (10) indicates that this would be the case. As stated previously, it would also be good to provide some intuition for why these step sizes make sense.

– There is a space missing between the numbers and “dB” on lines 268 and 271.

– What does it mean that “SNIPS does not handle general content images”?

– Several titles in the bibliography are improperly capitalized (probably missing some {} in the BibTeX file), including [3], [7], [8], [14], [21], [24], [30], [35], [36], [40], [48], [50], [53], [54], [55], [56], [57].

UPDATE:
After reading the other reviews and the author responses, I have decided to lower my score from 7 to 5 due to the lack of comparisons to established baselines, which make it difficult to establish the usefulness of the method.

**Time Spent Reviewing:**

3

---

> ### Author Response · Authors · 2021-08-09
> **Author Response to Reviewer d1T5**
>
> Thank you for the detailed review and thoughtful feedback. Below we address specific questions and comments.
>
> **Explained Derivations and the SVD:** Regarding the SVD, we made use of the regular (non-reduced) decomposition that outputs an orthogonal $\mathbf{V}$ which is $N \times N$. This convention is also standard (for instance, we used the PyTorch implementation for SVD), but it is less efficient than the reviewer’s suggestion (which is to use the reduced SVD). However, we cannot apply the reduced SVD here as the non-square matrix $\mathbf{V}$ would not be invertible, and Equation 4 would not hold.
> Equation 5 shows a rewrite of $\mathbf{y}$ in terms of $\mathbf{\tilde{x}}$, as requested. The reviewer is right in suggesting that it would be helpful to explain why $\mathbf{z}\_T - \mathbf{\Sigma n}\_T$ being white and independent of $\mathbf{\tilde{x}}\_T$ is desirable. We shall add a sentence explaining this in line 165. We will also add some explanations throughout the derivations, as much as the page limit allows us.
>
> **The requirement $\sigma\_i s\_j \neq \sigma\_0$**: We are sorry for this confusion - this requirement is almost meaningless, and was added only for ease of the derivations. Without it, we would have needed to include a special formula when the equality holds. Our tests (not included in the paper) show that the proposed algorithm works in such a scenario just as well, with no fear of instability. Since this has not been made clear in the paper, we will add a footnote clarifying this matter. We should note that the algorithm also works when the two terms are close but not equal – some of the experiments in the paper do have this property. If the reviewers find it necessary, we can add an appendix showing numerical stability experiments.
>
> **Notation choices:** We agree that the notation of $i\_1$ should be changed to $i\_j$. We will fix this in the camera-ready version of the paper. $i\_1$ (or $i\_j$) is defined only for $s\_j \neq 0$, and does not pose a problem for the case of $s\_j = 0$, since there is no correlation between $\mathbf{z}\_T$ and $\eta_{T,i}$ for all $i$ (as explained in line 185). This definition of the lack of correlation does not depend on a definition of $i\_1$, thus it is well defined. To avoid confusion: the definition between lines 182 and 183 is for $s\_j \neq 0$, and the definition on line 185 is for $s\_j=0$.
>
> **Choice of Annealed Noise:** The reviewer correctly points out that on line 176, we described the marginal distribution of $\eta\_i$. We will clarify this point in the camera-ready version and adopt the ideas proposed by the reviewer. Please note, however, that $\eta\_{T,i}$ are not scaled versions of $\mathbf{z}\_T$, but rather portions of it (i.e. $\mathbf{z}\_T$ is a sum of a subset of the $\eta\_{T,i}$’s). Therefore, we cannot directly construct $\eta\_{T,i}$ from $\mathbf{z}_T$, which explains our difficulties in writing this part of the paper. We will also add a brief motivation for Equation 6 in the camera-ready version.
>
> **Minor issues:** We will take care of each and every comment in the revised version. More specifically,
> \-	“the entries in the derived score switch mode” – We will better explain this part.
> \-	Line 130: we will modify the text and better explain the difficulties we refer to.
> \-	Mentioning the prior - we agree. We will add the notation that $\mathbf{x}$ is sampled from $p(\mathbf{x})$ in the beginning of section 3.1.
> \-	Superscript 2: We will move the footnote mark on line 144 to the end of the sentence.
> \-	White versus Heteroscedastic: The reviewer correctly points out our inaccurate use of the word “white”. We will fix this for the camera-ready version of the paper.
> \-	Section 3.1 and 3.2 notations: We will fix the text to be consistent with the notations regarding $\sigma\_i$ and $s\_j$.
> \-	The measurements in the Hessian approximation: We will add such an explanation.
> \-	Spaces: We will add a space between the numbers and dB.
> \-	Handling general content images: A limitation of the current version of SNIPS is that the experiments shown work on specific-class image datasets (such as celebrity faces, bedrooms, or towers). The algorithm cannot currently handle images with arbitrary real-world content. This limitation is of course tied to the denoiser network used, and not to the algorithm itself, but it is still a limitation.
> \-	References: We will carefully review the bibliography for the camera-ready version.

---

### Official Review · Reviewer_kSnH · 2021-07-16

**Rating:** 5
**Confidence:** 4

**Summary:**

The paper discusses an algorithm for sampling from the posterior distribution of a linear inverse problem with the white Gaussian noise, given the MMSE Gaussian denoiser.


---Edit after author responses---

After reading the reviews and responses I decided to decrease the score (from 7 to 5) mainly because of insufficient evaluation of the proposed method and issues with the derivations in Section 3.

**Limitations And Societal Impact:**

The authors discuss the limitations of the work.

**Main Review:**

The paper has a nice literature overview and is overall well written. The proposed method combines the newly proposed blurring of the samples by the additional noise with dependence on the measurement noise and the Langevin dynamic algorithm performed in the SVD domain of the known degradation matrix with the correction of the speed size. The sampling relies on an available pre-trained MMSE Gaussian denoiser. The experimental results demonstrate realistic sampling on image deblurring, super-resolution, and compressive sensing tasks.

Though the authors discuss the limitations of the proposed algorithm, in the numerical results the score function is estimated on the base of the same dataset, from which the linearly transformed and the blurred pictures come from. The resulting sampled pictures look really nice, but  which could be applications of the presented results? Would it be possible to assume a bit more realistic setting, when the source database is not available?


**Time Spent Reviewing:**

4

---

> ### Author Response · Authors · 2021-08-09
> **Author Response to Reviewer kSnH**
>
> Thank you for your review and thoughtful feedback. Below we address specific questions and comments.
>
> **Source Dataset:** First of all, we should clarify that our score function approximation is built on a training set of images, but the test images we experiment on are external to this set, so as to make these tests fair. As for applications, our algorithm can handle any inverse problem related to the families of images we have trained the MMSE denoisers on. Going beyond these sets and handling general content images is a major challenge, and we mention it as a critical future research direction in Section 6.

---

### Official Review · Reviewer_GwuL · 2021-07-17

**Rating:** 4
**Confidence:** 4

**Summary:**

The authors address the problem of conditional sampling using Langevin dynamics. It is known that annealed Langevin dynamics speeds up convergence, and prior work that uses Langevin dynamics for conditional sampling does not correctly model the effects of annealing. Specifically, since there is some annealing noise added to the estimate, the likelihood distribution of the measurements given an estimate will change, and the authors derive approximations to the correct functional form. Experiments show that their new estimator can be implemented in practice.


**Ethical Concerns:**

No ethical concerns.

**Limitations And Societal Impact:**

No societal concerns.

**Main Review:**

__Strengths__:
1. In theory, existing work cannot handle the challenges posed in the paper. The authors define their problem clearly, and they are able to derive approximations to the posterior distribution given access to a generative model that estimates annealed priors.

__Weaknesses__:

1. The key point of this paper is that the distribution $p(y|x)$ must be modified when you do annealing of the distribution $p(x)$, since this can introduce dependencies between the measurement noise and the annealing noise. Prior work would use Bayes rule to calculate the posterior distribution under annealing, and hence would get an incorrect functional form for the annealed posterior distribution. Is the error in your approximated distribution smaller in some helpful way? Overall, since there are no error bounds for sampling from annealed posteriors that converge to the true posterior, I fail to see the significance of an exact derivation of the true annealed posterior. In fact your derivation only gets an approximation to the annealed posterior, and hence will introduce its own error.

1. The writing is extremely imprecise and the contributions are unclear.  Other than statements that prior work cannot handle ill-posed operators and noise, there isn't any empirical or theoretical evidence that prior work fails or that the proposed approach fixes this problem.

1. The authors say that the rational behind sampling in the SVD domain is in the appendix, but this only contains derivations for the conditional likelihood in the SVD domain. There is no theoretical justification for sampling in the SVD domain.

1. There are literally no comparisons to baselines. Reference [1] shows that a direction application of Bayes' rule gives good empirical results for inpainting and compressed sensing, while prior work cited in the paper has shown it for super-resolution, denoising, etc. Does your method have any empirical advantage?

__Missing references__:

[1] shows that posterior sampling is optimal for compressed sensing and needs to be cited.

[1] Jalal, Ajil, Sushrut Karmalkar, Alexandros G. Dimakis, and Eric Price. "Instance-Optimal Compressed Sensing via Posterior Sampling." arXiv preprint arXiv:2106.11438 (2021).

**Time Spent Reviewing:**

3

---

> ### Author Response · Authors · 2021-08-09
> **Author Response to Reviewer GwuL**
>
> Thank you for this review and thoughtful feedback. In the following we address specific questions and comments.
>
> **Approximation Error:** When referring to $p(\mathbf{y}|\mathbf{x})$ in this comment, we assume the reviewer meant $p(\mathbf{y} | \mathbf{\tilde{x}})$, which is what the annealed Langevin leads to. And yes, it is true - the dependencies between the measurements and annealing noise pose a challenge for deriving this expression, and hit a dead-end when treated in the pixel domain, as shown in Appendix C. Our derivation relies on SVD for decoupling the measurements equation, and an intricate connection we pose between these two noises. Yes, our approach also uses an approximation (which is assuming no statistical dependencies between transformed samples in the synthesized image and the null-space of $\mathbf{H}$). However, when it comes to handling of general noisy inverse problems, there is no alternative in the literature to compare with. Prior work that could have used the Bayes rule considers noiseless problems (e.g. [18]), or treat simplified tasks such as denoising and inpainting (e.g. [19]). Thus, the question of comparing approximation errors becomes irrelevant. As such, the contribution of our paper is in deriving the required posteriors and showing their applicability to several inverse problems. We agree that a bound on our approximation error would be beneficial, and this is left for future work.
>
> **Relation to Prior Work:** The comment states “there isn't any empirical or theoretical evidence that prior work fails or that the proposed approach fixes this problem”. The related work relevant to this comment are those papers (see section 2) that assume no noise in their measurements and/or a limited set of operators $\mathbf{H}$ that have all ‘1’-s or ‘0’-s singular values. Clearly, our work goes beyond these assumptions, allowing measurements noise and **any** operator $\mathbf{H}$. Another branch of earlier work relevant to this comment considers denoising and inpainting, (e.g. [19]), but again this work cannot be extended easily for a general $\mathbf{H}$ degradation. So, be it theory or empirical evidence, the proposed paper offers a significant step forward with respect to prior work, and the exposition of this is given in a very clear way.
>
> **Sampling in the SVD Domain:** The justification in the appendix shows why the derivation of the score function works in the SVD domain, but does not work in the original image domain. As for theoretical justification for sampling in the SVD domain, please refer to Equation 4 and lines 157-159.
>
> **Baseline Comparisons:** Our method presents a unified framework for all noisy linear inverse problems as posed in Equation 1. This is the main advantage of our method over previous works (such as [19, 43]) that had to derive specific algorithms for each and every inverse problem. Reference [1] that the reviewer presents was published on ArXiv on June 21st, 2021. It is unrealistic to expect us to cite them when the NeurIPS submission deadline was May 28th, 2021.
> Nevertheless, in order to comply with this important comment, we will add comparisons of our method, its MMSE estimate (averaging of multiple samples), PnP [49], and RED [36], all relying on the same denoiser. We will evaluate these solvers using PSNR as well as LPIPS [1r] for perceptual quality.
>
> **References:**
> [1r] Zhang, Richard, et al. "The unreasonable effectiveness of deep features as a perceptual metric." *Proceedings of the IEEE/CVF Conference on Computer Vision and Pattern Recognition.* 2018.

---

> > ### Comment · Reviewer_GwuL · 2021-09-07
> > **Follow up to author response**
> >
> > Thank you for your response. I have read the other reviews and your responses to the reviews. However, I still have concerns.
> >
> > **Baseline comparisons**: I apologize for the confusion, I wasn't asking for a comparison to [2]. There is no known failure for straightfowardly extending [1] to noisy inverse problems, and [2] shows that even a crude approximation in the pixel domain will give good experimental results. This submission and [2] use the same NCSNv2 prior, and it isn't clear whether the SVD techniques help in any way.
> >
> > Prior work cited in this submission can handle image deblurring, and this submission should provide a comparison ( or if it is the same algorithm, this should be explicitly stated). Similarly, prior work for noiseless super-resolution & compressed sensing (which have been cited in this submission) can still be run in the setting where there's noise in the measurements, and this submission should provide a comparison. Sure, the prior work may be running their algorithms on an incorrect version of the posterior distribution, but this comparison is crucial in order to understand whether the proposed techniques have any benefit -- perhaps the NCSNv2 model used is robust to errors in the posterior distribution, and there's no need for doing an SVD decomposition.
> >
> > The comparisons to RED and PnP will be meaningful, but should have been included in the main submission.
> >
> > In my opinion, the theoretical results alone don't meet the threshold for NeurIPS, and the absence of any experimental baselines is a major concern. Hence I am not increasing my score.
> >
> > **(Minor concern) Missing references**: This is a minor point, but this submission entirely ignores prior work that also solves inverse problems using trained & untrained generative priors [2-7]. The submission continually refers to denoisers, but the experiments use an NCSNv2 model, which achieves state-of-the-art scores for generative modelling. Hence referring to it as a denoiser isn't accurate, as it is far more powerful and more complicated than traditional denoisers.
> >
> > [1] Y. Song and S. Ermon. Generative modeling by estimating gradients of the data distribution. In Advances in Neural Information Processing Systems, pages 11918–11930, 2019.
> >
> > [2] Jalal, Ajil, Sushrut Karmalkar, Alexandros G. Dimakis, and Eric Price. "Instance-Optimal Compressed Sensing via Posterior Sampling." arXiv preprint arXiv:2106.11438 (2021).
> >
> > [3] Bora, Ashish, Ajil Jalal, Eric Price, and Alexandros G. Dimakis. "Compressed sensing using generative models." In International Conference on Machine Learning, pp. 537-546. PMLR, 2017
> >
> > [4] Hand, Paul, Oscar Leong, and Vladislav Voroninski. "Phase retrieval under a generative prior." In Proceedings of the 32nd International Conference on Neural Information Processing Systems, pp. 9154-9164. 2018.
> >
> > [5] Hand, Paul, and Vladislav Voroninski. "Global guarantees for enforcing deep generative priors by empirical risk." In Conference On Learning Theory, pp. 970-978. PMLR, 2018.
> >
> > [6] Heckel, Reinhard, and Paul Hand. "Deep decoder: Concise image representations from untrained non-convolutional networks." arXiv preprint arXiv:1810.03982 (2018).
> >
> > [7] Heckel, Reinhard, and Mahdi Soltanolkotabi. "Compressive sensing with un-trained neural networks: Gradient descent finds a smooth approximation." In International Conference on Machine Learning, pp. 4149-4158. PMLR, 2020.

---

### Official Review · Reviewer_VyNV · 2021-07-23

**Rating:** 4
**Confidence:** 4

**Summary:**

This paper proposes an algorithm called SNIPS that solves noisy linear inverse problems via an approximate MMSE estimator which makes use of pre-trained Gaussian denoisers. The estimator is based on MCMC sampling which is driven by a synthetically annealed version of Langevin dynamics. The motivations and background leading up to the algorithms derivation are well discussed. Experiments show interesting results with regard to the diversity and faithfulness of the images generated.


---Edit after author responses---

Having read the other reviews and the responses posted by the authors, I am reducing my original scores from 6 --> 4.
My main issues are that the submission lacks (i) a comparison with an experimental baseline, and (ii) clarity in explaining the technical sections of the paper.

**Limitations And Societal Impact:**

The limitations are adequately addressed.

**Main Review:**

Sections 1 and 2 are well written. The problem is clearly stated and the method is well motivated with a good discussion on prior work. However, the authors do not clearly state what the contributions of the current paper are. I suggest the authors highlight these in a subsection at the end of Section 2. It appears Section 3 and 4 are the main contributions of this work, but they fall short in delivering the clarity desired from any original piece of work.


Section 3 is technical and intricate but lacks clarity throughout.
It is not clear why access to the true score function \nabla log p(x|y) is not available. The assumptions that lead to these limitations need to be made clearer at the beginning of Section 3.
The derivations in Section 3.1 are not explained well. It makes very little sense to the reader as to why the specific noise levels and correlations are chosen. The authors ought to provide a lot of intuition here. In Section 3 it is very unclear what quantities are known for the problem, which of those are assumed to be known for the purpose of the analysis, and which can be assumed to be known while designing the algorithm.

Section 2 makes a lot of notational assumptions which are very hard to follow.

While the derivations in Section 3.2 are pushed to the appendix, the authors need to remind the reader what the significance of each of the equations 7-11 is by mentioning in words what these equations mean.
In Algorithm 1 the notation of the dagger and >, /> appear non-standard. I urge the authors to add a caption to the algorithm which clarifies this notation.
Line 111: Is the algorithm in equation (2) called the Unadjusted Langevin algorithm (ULA)? If so perhaps the authors should mention that name.



Additional comments/typos:
Line 47: GAN-based and Langevin sampling --> based on GANs and Langevin sampling
Line 182: The quantity \eta_T is not defined explicitly. Please do so
Notation issue: \eta_T, n_T are defined as V^T \eta and V^T n respectively.
\eta_T,i and n_T,i however is a component of \Sigma V^T\eta_i and \Sigma V^T n_i which is inconsistent. Please change the notation to make it easier to read.
Equations following line 182, please remind the reader that i_1 is a function of j.

**Time Spent Reviewing:**

3

---

> ### Author Response · Authors · 2021-08-09
> **Author Response to Reviewer VyNV**
>
> Thank you for the detailed review and thoughtful feedback. Below we address specific questions and comments.
>
> **Contributions:** Please note that the main contributions of this work are summarized and highlighted at the end of section 1, lines 96 through 103.
>
> **Clarity of Section 3:** Section 3 is indeed very technical. We welcome any suggestions that can help improve its clarity, and we will make a sincere effort to improve it.
> More specifically, the true score function is unavailable for a number of reasons, as explained in previous work [41]. Most prominently, it is not well defined when $p(\mathbf{x})=0$ (the log of zero is not defined). The migration to the blurred score function is part of the desire to move from regular Langevin dynamics (that requires many thousands of iterations) to the faster annealed version, and this has been thoroughly explained in earlier work as well, e.g.  [41]. The blurred posterior distribution $p(\mathbf{\tilde{x}}|\mathbf{y})$ cannot be dealt with directly in the pixel domain, as clarified in Appendix C, thus resorting to a decoupling via SVD. This appendix also explains the choice we have made on the inter-connection between the measurements and the annealed noise vectors.
> The known quantities are stated at the beginning of section 3 in line 144, and the description of Algorithm 1 states the necessary inputs. We will go carefully through these parts to make sure that they are complete and detailed.
>
> **Clarity of Section 3.2:** We will scan this section for better defining our derivations and notations, as suggested. As for line 111, indeed, this is known as either “Langevin dynamics” or “Unadjusted Langevin Algorithm”. We stick to the “Langevin dynamics” name, as used in [41].
>
> **Notation Issues:** The reviewer correctly points out a problem: $\mathbf{n}\_T$ is defined as $\mathbf{V}^T \mathbf{n}$, while $n\_T$ is defined as an entry of $\mathbf{\Sigma V}^T \mathbf{n}$. This is indeed confusing, and it will be fixed for the camera-ready version by changing one of the notations. Notice that the problem does not appear for $\eta\_{T,i}$, but only $n\_T$. Notice that $\eta\_T$ (a notation which we did not use in the paper) is equivalent to $\eta\_{T,i}$. The index $i$ denotes the current step in the Langevin process, and not an entry in a vector. The reviewer points out that we should remind the reader that $i\_1$ is dependent on $j$. We agree and we will add such a reminder.

---

### Decision · Program_Chairs · 2021-09-28

**Decision:**

Accept (Poster)

**Comment:**

This paper solves linear inverse problems via an approximate MMSE estimator using pre-trained Gaussian denoisers. The results look very good since the paper uses NCSNv2 which is an excellent generative model.

The two major problems are:
There are really no comparisons with previous experimental baselines.
This is a major limitation for any scientific paper and the authors need to strengthen their evaluation section by discussing how prior work performs for the problems they solve. There are dozens of papers that solve deblurring, inpainting and compressed sensing problems using deep generative models. (Many papers use NCSNv2 and the more updated Score_SDE by Song et al. for various inverse problems and many others use other types of deep generators).

Typically visual comparisons and comparisons in MSE and SSIM would help the readers understand when the current method outperforms previous work.

Further, the technical contribution of the paper was actually not clear to the reviewers. We would encourage the authors to clarify their presentation based on the reviewer feedback and revise their manuscript.


**Consistency Experiment:**

NeurIPS has a long history of experimentation. In 2014, NeurIPS ran an experiment in which 10% of submissions were reviewed by two independent committees to quantify the randomness in the review process. This year, we repeated a variant of this experiment to see how the quality of the review process has changed over time.  This paper was part of the experiment and was therefore assigned to two committees (consisting of reviewers, an Area Chair, and a Senior Area Chair) that reached independent decisions.  If both committees made the same recommendation, this recommendation was followed. If a single committee recommended acceptance, the paper was accepted (with the exception of a few cases in which the other committee identified what we considered a fatal flaw, e.g., an error in a key result).

This copy’s committee reached the following decision: **Reject**

The other committee assigned to the paper recommended **Accept (Spotlight)**.  You can find the other set of reviews, along with any follow up discussion with the authors here:
https://openreview.net/forum?id=pBKOx_dxYAN